# Mercury: A Code Efficiency Benchmark for Code Large Language Models

**Mingzhe Du**[1,2], **Luu Anh Tuan**[1], **Bin Ji**[2], **Qian Liu**[3], **See-Kiong Ng**[2]
[1]Nanyang Technological University
[2]National University of Singapore
[3]Sea AI Lab
{mingzhe001, anhtuan.luu}@ntu.edu.sg, {jibin, seekiong}@nus.edu.sg, liuqian@sea.com

## Abstract

Amidst the recent strides in evaluating Large Language Models for Code (Code LLMs), existing benchmarks have mainly focused on the functional correctness of generated code, neglecting the importance of their computational efficiency. To fill the gap, we present *Mercury*, the first code efficiency benchmark for Code LLMs. It comprises 1,889 Python tasks, each accompanied by adequate solutions that serve as real-world efficiency baselines, enabling a comprehensive analysis of the runtime distribution. Based on the distribution, we introduce a new metric `Beyond`, which computes a runtime-percentile-weighted `Pass` score to reflect functional correctness and code efficiency simultaneously. On *Mercury*, leading Code LLMs can achieve 65% on `Pass`, while less than 50% on `Beyond`. Given that an ideal `Beyond` score would be aligned with the `Pass` score, it indicates that while Code LLMs exhibit impressive capabilities in generating functionally correct code, there remains a notable gap in their efficiency. Finally, our empirical experiments reveal that Direct Preference Optimization (DPO) serves as a robust baseline for enhancing code efficiency compared with Supervised Fine Tuning (SFT), which paves a promising avenue for future exploration of efficient code generation. [1]

## 1 Introduction

The domain of code generation, which aims to empower computers to autonomously generate code based on natural language task descriptions (NL2Code), has long been considered a promising way to facilitate interaction between humans and computers [51, 49]. The recent emergence of Large Language Models (LLMs) has spurred a new wave of NL2Code models [38, 42, 14, 33, 4], which leverage the impressive language understanding and generative capabilities of LLMs to drive forward the ambitious goal of synthesizing high-quality code from natural language instructions.

To measure the quality of code, recent code generation benchmarks mainly focus on evaluating their functional correctness via test case fuzzing [31]. This approach assesses the outcome congruence between the LLM-generated and canonical solutions by executing bespoke test cases. For instance, HumanEval [9] and MBPP [3] collected a small but fine set of handcrafted tasks with test cases. EvalPlus [30] further consolidates these two above benchmarks by augmenting the case scope. On the contrary, APPS [18] widely gathered over 5,000 public coding tasks from online platforms. Despite these strides, there is a discernible oversight in current code generation benchmarks concerning the code efficiency evaluation, although that is critical in software development [50, 52]. Moreover, handcrafting diverse solutions and test cases to cover all scenarios is infeasible [30]. In light of these findings, we highlight vital limitations inherent in the existing code generation benchmarks:

---

[1]Our code and data are available on GitHub: https://github.com/Elfsong/Mercury.

38th Conference on Neural Information Processing Systems (NeurIPS 2024) Track on Datasets and Benchmarks.

Given an array of integers `nums`, sort the array in ascending order and return it.

```python
# Solution A
def sortArray(self, nums):
    i = 0
    while i < len(nums)-1:
        j = i + 1
        while j < len(nums):
            if nums[i] > nums[j]:
                nums[i],nums[j] = nums[j], nums[i]
            j += 1
        i += 1
    return nums
```

Runtime
**5714 ms** 🐢

```python
# Solution B
def sortArray(self, nums):
    def quicksort(nums, l, r):
        if r - l ≤ 1: return
        # Function partition not shown for clarity
        pivot = partition(nums, l, r)
        quicksort(nums, l, pivot)
        quicksort(nums, pivot+1, r)
    quicksort(nums, 0, len(nums))
    return nums
```

Runtime
**121 ms** 🐇

Functional Correctness: **Passed** ✅
Computational Efficiency:   **Slow** 🙁

Functional Correctness: **Passed** ✅
Computational Efficiency:   **Fast** 🙂

Figure 1: Executing these two LLM-generated codes on 100 test cases. While both codes successfully follow the task instruction and pass all test cases, the *right* snippet notably excels in code efficiency, completing in a mere 121 ms compared to the 5,714 ms consumed by the *left* snippet. As Code LLMs become widely used in the real world, code efficiency determines factual productivity, where Mercury can gauge the vital metric.

1. **Absence of Code Efficiency Evaluation.** Existing code generation benchmarks focus on assessing functional correctness while overlooking the evaluation of code efficiency [9, 3, 18]. As illustrated in Figure 1, despite both code snippets can handle the sorting task functionally, the *right* efficient solution (*121 ms*) is nearly 50 times faster than the *left* inefficient solution (*5,714 ms*). This striking runtime differentiation underscores the necessity of incorporating code efficiency assessments within code generation benchmarks, encouraging Code LLMs to produce not only correct but also efficient code.

2. **Insufficient Test Case Coverage.** As shown in Table 1, most code generation benchmarks manually build a small number of test cases or extract the accompanying test cases from existing resources, potentially overlooking edge cases and nuanced code behaviors [9, 3]. For example, Figure 8 displays that *HumanEval #55* contains only 3 test cases, testing up to the 12th Fibonacci number [9]. Its given canonical solution will quickly reach the recursion depth limitation when computing a larger Fibonacci number (the recursion limitation depends on the environment). Therefore, notwithstanding the generated code satisfies all test cases, such success does not necessarily equate to assurance of functional correctness and much less to code efficiency.

3. **Lack of Task Diversity.** Another noticeable deficit of existing code generation benchmarks is the insufficient diversity and complexity in their tasks [9, 3, 31]. Since most benchmarks only consist of elementary-level programming tasks, recent Code LLMs can effortlessly tackle most tasks regardless of their actual capacities [52]. This flaw results in these benchmarks failing to pose a substantial challenge to Code LLMs and truly reflect their underlying potential.

**Code Efficiency.** *Code efficiency refers to the performance measure of time and space complexity to accomplish a specific task.* Efficient code can improve user experience, save energy, and make applications more sustainable and cost-effective. Compared with the scalable memory space, execution time is the performance bottleneck of most codes. Consequently, this work focuses on the time dimension of code efficiency.

**Our Benchmark.** In this work, we introduce *Mercury*, a novel code generation benchmark designed to assess and improve the code efficiency of Code LLMs. As depicted in Figure 2, *Mercury* comprises 1,889 Python programming tasks with three difficulty stratification, which is divided into two datasets for model evaluation and fine-tuning separately. For each evaluation task, we assign a test case generator to remedy the shortfall of test case coverage. In measuring code efficiency, the primary challenge stems from normalizing the absolute runtime across tasks that have diverse runtime ranges. Thus, we collect and locally execute numerous historical solutions for each task to form a runtime distribution and leverage the runtime percentile of LLM-generated code on the distribution instead of the absolute runtime to evaluate code efficiency. Furthermore, to mitigate performance discrepancies attributed to irrelevant processes and diverse hardware configurations, we set up an isolated sandbox environment for task execution to establish local runtime distributions.

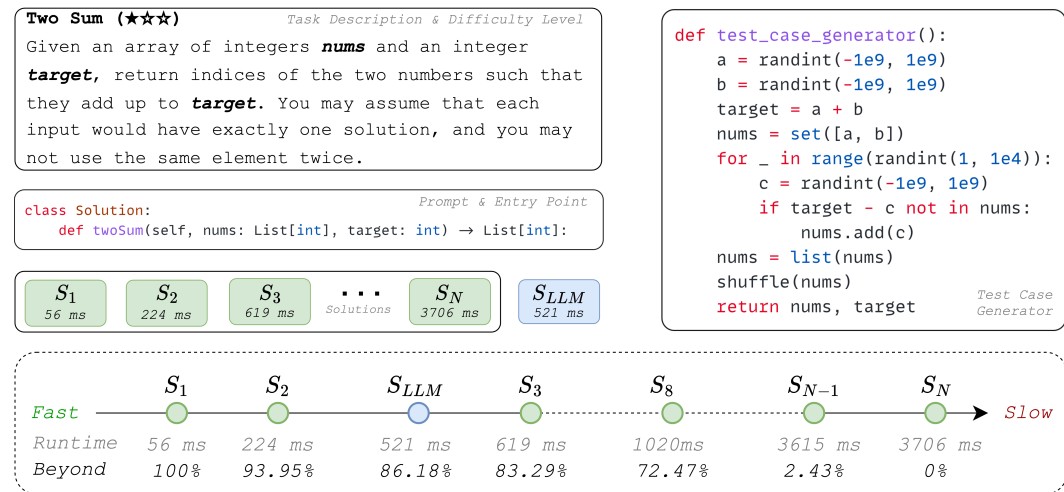

Figure 2: An overview of *Mercury* dataset. Each *Mercury* task has a task description, a test case generator, a prompt & entry point, and corresponding solutions. To evaluate code efficiency, we introduce the Beyond metric, which signifies the runtime percentile of the LLM-generated code on the runtime distribution supported by corresponding solutions. In this example, the LLM-generated code executes in 521 ms, outpacing 86.18% of collected solutions on the runtime distribution. Consequently, the Beyond metric in this case is 86.18%.

**Contribution.** Our work aimed to fill the code efficiency evaluation gap in code generation benchmarks with the following key contributions:

- **Dataset.** We collect a novel code generation dataset *Mercury* designed to assess and improve Code LLM code efficiency in Section 2, accompanied by an extensible open-source data collection framework for enriching *Mercury* with more tasks and programming languages.
- **Metric.** We propose the first efficiency-focused code generation metric Beyond and establish a benchmark to evaluate leading Code LLMs using this metric in Section 3.
- **Baselines.** In Section 4, we detail our extensive analysis of two baselines to enhance code efficiency while maintaining functional correctness. Experiment results reveal that despite Code LLMs excelling in functional correctness, there is still considerable potential to elevate efficiency.

Table 1: A comparison of *Mercury* to existing NL2Code benchmarks. *Mercury* distinguishes itself by including a set of distilled high-quality solutions and a dedicated test case generator for each task. * signifies that the solution number can be further expanded by the data collection framework.

| Benchmarks | Tasks | Sources | Cases | Solutions | Difficulty | Efficiency |
|---|---|---|---|---|---|---|
| HumanEval | 164 | Crowd Source | 8.08 | 1 | 1 | ✗ |
| MBPP | 257 | Crowd Source | 3.01 | 1 | 1 | ✗ |
| APPS | 5,000 | Online | 21.2 | 23.4 | 3 | ✗ |
| **Mercury** | 256 | Online + Filters | $+\infty$ | 18.4 * | 3 | ✓ |

## 2 Mercury Datasets

We initiate this work by collecting public programming tasks on Leetcode [27]. Subjecting these questions to a series of filters, we distilled them down to 1,889 high-quality tasks. A difficulty-balanced subset of 256 tasks was randomly selected to form the **Mercury-eval** benchmark, which obtains an average of 18.4 solutions for each problem. The remaining tasks have been designated as the **Mercury-train** dataset for baseline training (detailed data distribution is listed in Appendix Table 6). To enhance clarity within this paper, we employ *Mercury* to denote **Mercury-eval** unless otherwise specified.

**Data Schema.** As illustrated in Figure 2, *Mercury* offers a unified data schema to streamline the evaluation procedure and bolster further development endeavors. The data scheme encompasses these principal components: (1) **Task Description** contains the task instruction interpreted into a plain text format, along with illustrative examples and constraints of inputs and outputs. (2) **Test Case Generator** refers to a Python code snippet designed to automatically produce a comprehensive set of test cases in accordance with the specifications laid out in the task description. (3) **Solutions** are sampled from Leetcode historical submissions. Each solution within Mercury has undergone rigorous testing, and Locality-Sensitive Hashing [21] is employed to prevent the inclusion of any identical solutions. (4) **Prompts and Entry Points** where prompts act as the initiating prefixes for LLM code generation and entry points denote the start point for code execution. We delineate the definition of *Mercury* fields in the Appendix Table 5.

**Task Filters.** *Mercury* tasks originate from public programming problems on Leetcode. To assure the quality and uniformity of the dataset, we distilled gathered tasks based on the following conditions:

1. **Number of Solutions.** To establish a solution runtime distribution for each task, we filtered out tasks having less than two associated solutions. After excluding these tasks, *Mercury* tasks possess an average of 18.4 unique solutions.
2. **Restricted Data Structure.** Above the inherent Python data types, *Mercury* also incorporates two custom data types: Binary Tree and Linked List (the specific structure definitions can be found in Appendix Figure 4), which increases *Mercury's* diversity and escalates its difficulty level. Tasks that contain other data structures will be removed.
3. **Unique Outputs.** Certain Leetcode tasks may permit non-unique answers. For example, a result list can be returned in any order. Evaluating all possible answers can drastically complicate the test case verification process. To eliminate this problem, we harness the corresponding test case generator to generate $N$ test cases $T_i = \langle Input_i, Output_i \rangle$ $s.t.$ $i \in \{0, 1, \cdots, N\}$ and execute $T$ on different solutions $S_m$ $s.t.$ $m \in \{0, 1, \cdots, M\}$ to observe if all $Output_i = S_m(Input_i)$ $s.t.$ $i \in \{0, 1, \cdots, N\}$ remain identical. Any tasks that potentially yield non-unique answers were subsequently excluded.

**Task Difficulty.** Most existing NL2Code benchmarks predominantly comprise simplistic tasks, leading to a situation where LLMs of varied capabilities address most tasks effortlessly and yield indistinguishable high scores [52, 22]. To alleviate this issue, *Mercury* inherits the difficulty categorization from Leetcode, *i.e.*, Easy, Medium, and Hard. The stratification aims to probe the upper bounds of Code LLM capabilities, delivering a more evident distinction between various Code LLMs.

**Test Case Generator.** Manual creation of test cases can be a laborious process. To gather sufficient test cases to conduct an exhaustive assessment, we assign a test case generator for each evaluation task, which can produce a full range of test cases to thoroughly evaluate the functional correctness and code efficiency of given solutions. Specifically, We feed $pretty\_content$ into GPT-4 [38] to generate an initial test case generator snippet. To confirm the effectiveness of the initial generator, we subsequently create 24 test cases by the generator and submit these cases to the Leetcode Online Judge (OJ) system. Should any of the generated test cases not pass the LeetCode OJ validation, we manually revise the generator until all generated cases can be successfully validated.

## 3   Code Efficiency Metric

In the domain of software development, code efficiency can be defined as the absolute code runtime for executing a given test case set [8]. Nonetheless, a primary obstacle in benchmarking code efficiency is normalizing runtime measurements across disparate environments. For instance, a sub-optimal solution might have a faster absolute runtime on high-performance hardware than an optimal solution on low-performance hardware. Moreover, different operation systems and code interpreters may also fluctuate the code runtime. Therefore, absolute runtime fails as a consistent and reliable code efficiency benchmark metric. To address this issue, an intuitive approach involves modeling a devoted runtime distribution for each task and calculating the average runtime percentiles of LLM solution samples over the runtime distribution. With this idea in mind, we proposed a normalized code efficiency metric Beyond:

$$p_k^n = \frac{max(R^n) - clip(r_k^n, min(R^n), max(R^n))}{max(R^n) - min(R^n)}, \qquad Beyond = \frac{\sum_{N,K}^{n=0,k=0} p_k^n}{N \cdot K}. \qquad (1)$$

Where $N$ is the total number of tasks, and $K$ denotes the size of LLM solution samples. For a specific task $n \in N$, $R^n$ is the runtime array corresponding to the collected historical solutions, and $r_k^n$ s.t. $k \in K$ denotes the runtime for the $k$-th LLM solution. $clip$ is a function to constraint the value $r_k^n$ in the range $[min(R^n), max(R^n)]$. *Runtime* is defined as the period from the solution instantiation to the evaluation across all test cases, culminating with a successful termination (More engineering details can be found in Appendix Section A.3). Since any case failure of the $k$-th solution results in $r_k^n \to +\infty$ and then $p_k^n = 0$, Beyond can reflect functional correctness as well.

**Untrusted Code Execution.** Since most Code LLMs are trained on an extensive code corpus from unverified sources, there is an intrinsic risk that these models may produce malicious code when driven by specific meticulous prompts [9]. The direct execution of synthesized code raises significant security concerns. To alleviate the risk of running untrusted code, we engage a robust sandbox to execute code in an isolated environment. Sandbox details are deliberated in Appendix A.3

**Environment-agnostic Evaluation.** To ensure fair comparison across diverse configurations, we run each task $n$ with corresponding test cases locally and aggregate their runtimes into the runtime array $R^n$. Appendix Figure 10 illustrates the Beyond score of two LLMs ('deepseek-coder-33b' and 'deepseek-coder-6.7b') over three distinct hardware specifications: the micro-tier (0.25 CPU cores), the small-tier (0.5 CPU cores), and the standard-tier (1 CPU core). The results demonstrate that *Beyond* remains consistent over different hardware configurations.

## 4 Experiments

In this section, we present a series of baseline experiments to improve code efficiency by training on *Mercury-train* dataset and assessing on the *Mercury-eval* dataset. Our empirical study encompasses 10 open-source LLMs with a broad parameter spectrum from 1.3 to 34 billion. For each LLM, we compare the performance of the original model and two optimization strategies, Supervised Fine-Tuning (SFT) and Direct Preference Optimization (DPO), for their potential to optimize LLM generating functionally correct and computationally efficient code. Finally, we analyzed the underlying factors contributing to the failure of LLMs on the *Mercury-eval* dataset.

### 4.1 Baselines

**Supervised Fine-Tunning (SFT).** Within the SFT [6] method, an LLM undergoes additional training on a small dataset, which aims to specialize the LLM to perform better on certain tasks correlated to the training dataset. To optimize the code efficiency performance of Code LLMs, the most intuitive strategy is to fine-tune the Code LLM using optimal runtime solutions. In our experimental setup, we apply a unified prompt template for each Code LLM to ensure a fair comparison. The "pretty_content" attribute fills the **<task_content>** placeholder, the "prompt" attribute fills the **<code_starter>** placeholder, and the **<code_completion>** placeholder is completed with the fastest solutions. To steer Code LLMs towards generating the intended code completion format, we prepend a one-shot example to the prompt template. Appendix Figure 9 presents the generation template.

**Direct Preference Optimization (DPO).** Although SFT exemplifies a straightforward approach, it is susceptible to the pitfall of catastrophic forgetting [24]. To enable LLMs to align with human preferences while preserving their functional capabilities, existing methodologies employ reinforcement learning with human preference feedback (RLHF). However, RLHF introduces additional model complexities and potential instabilities, necessitating significant computing resources and extra reward model training [54, 5, 45]. DPO [40] bypasses these challenges by explicitly mapping reward functions and the optimal objective. This connection demonstrates that maximizing rewards under specific constraints can be effectively addressed through a singular training phase based on data reflecting human preferences. The DPO training procedure is elaborated in Appendix Section A.4.

### 4.2 Functional Correctness Benchmarks

**HumanEval** assesses the functional correctness of synthesized code derived from docstrings. It contains 164 distinct Python tasks that cover several programming areas, such as language comprehension, algorithm development, and simple mathematics [9]. **MBPP** has a sanitized collection of 257 entry-level Python programming problems. Each problem in this dataset consists of three components: a task description, an associated code solution, and three automated test cases to validate the code functionality [3]. Both HumanEval and MBPP harness the metric Pass to measure the Code LLMs' functional correctness, where a task is considered solved if the given solution passes all test cases, and the total fraction of solved tasks is reported as $Pass = N_{solved}/N_{total}$ [25].

### 4.3 Experimental Setups

**Configuration.** We employ LoRA [19] for both SFT and DPO experiments. We set $lora\_alpha = 16$, $lora\_dropout = 0.05$, and $lora\_r = 8$. The optimizer is *Adamw* [32], and the learning rate is $1e$-4 and $5e$-5 for SFT and DPO, respectively. For SFT experiments, we train each model in 200 steps. For DPO experiments, we set $\beta = 0.1$ and $training\_step = 500$. For code generation, we set the temperature as 0.2. For the *Beyond* metric calculation, we set $K = 5$. All experiments are conducted on two A100-80G GPUs. We employed Accelerate [17] for distributed training, DeepSpeed [1] for gradient partitioning, and BitsandBytes [15] for model quantization.

**Training Data.** We use *Mercury-train* for model training. As for the SFT process, we nominate the fastest solution as the supervised label, then format the training data as $\langle pretty\_content, prompt, solution\_optimal \rangle$. Regarding the DPO procedure, we select the top 5 pairs of solutions that exhibit the most significant discrepancy in runtime. The training date format is $\langle pretty\_content, prompt, solution\_fast, solution\_slow \rangle$.

### 4.4 Empirical Results

Functional correctness is the prerequisite for evaluating code efficiency for the code generation task. Our primary objective is to enhance the code efficiency without compromising the functional correctness. To this end, we first introduce the existing metric `Pass` to gauge the functional correctness [9] and then leverage `Beyond` to provide a holistic evaluation, encompassing both code efficiency and functional correctness. Finally, we measure the `Gap` between `Beyond` and `Pass` to reflect the baseline ability of improving efficiency while preserving correctness. These experiments aim to investigate the innate capabilities of cutting-edge Code LLMs and their potential after baseline fine-tuning. Therefore, extensive parameter optimization and prompt engineering were not pursued for the Pareto front. To deliver a comprehensive evaluation, we have further integrated the HumanEval and MBPP benchmarks as supplementary measures for appraising functional correctness [9, 3].

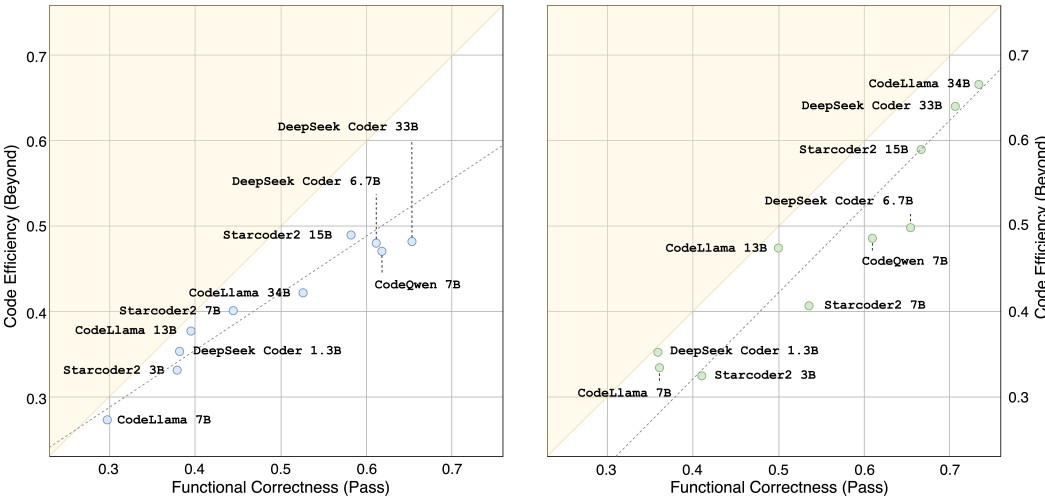

Figure 3: The horizontal axis represents the score for functional correctness, while the vertical axis indicates the score for code efficiency. The diagonal line represents perfect efficiency to the corresponding correctness. Points closer to this line indicate better efficiency improvements relative to their correctness. The *left* figure illustrates the performance of the baseline model, whereas the *right* one depicts the performance after DPO tuning.

**Functional Correctness.** Table 2 lists `Pass` scores over various Code LLMs, showing that larger models tend to provide better functional correctness. Except for the smallest model "deepseek-coder-1.3b-base", DPO invariably enhances the overall `Pass` scores across most Code LLMs, while SFT diminishes functional correctness on the largest two Code LLMs. These findings suggest that smaller models may struggle to integrate new knowledge while preserving their original functionality, and SFT may induce catastrophic forgetting in the pursuit of heightened code efficiency. Moreover, it is evident on *Mercury* that `Pass` scores of each model consistently decline as the difficulty level increases, indicating that the *Mercury* difficulty stratification is effective at probing the upper limitation of each Code LLM compared to the auxiliary benchmarks.

Table 2: Functional correctness (Pass) evaluation results. The underlined values denote the top-performed approaches among the original model and baselines. **The bolded values** denote the best performance on each benchmark. We sample one solution for each task to calculate *pass* score.

| Model Name | HumanEval | MBPP | Mercury | | | Overall |
|---|---|---|---|---|---|---|
| | | | Easy | Medium | Hard | |
| **deepseek-coder-1.3b-base** | 28.7 | 55.4 | 60.7 | 52.8 | 23.2 | 38.1 |
| + SFT | 24.2 | 46.2 | 58.9 | 53.6 | 25.3 | 36.2 (-1.9) |
| + DPO | 29.1 | 50.2 | 61.4 | 53.6 | 20.0 | 35.9 (-2.2) |
| **starcoder2-3b** | 31.7 | 57.4 | 56.1 | 52.1 | 21.6 | 37.8 |
| + SFT | 29.0 | 47.2 | 60.7 | 58.8 | 25.3 | 38.8 (+1.0) |
| + DPO | 33.5 | 59.6 | 62.5 | 61.0 | 23.4 | 41.1 (+3.3) |
| **deepseek-coder-6.7b-base** | 47.6 | 70.2 | 69.3 | 68.9 | 56.1 | 61.0 |
| + SFT | 56.1 | 59.6 | 69.1 | 71.4 | 57.7 | 62.2 (+1.2) |
| + DPO | 54.3 | 72.8 | 74.1 | 72.6 | 58.9 | 65.4 (+4.4) |
| **starcoder2-7b** | 35.2 | 54.4 | 63.6 | 61.7 | 29.2 | 44.3 |
| + SFT | 42.9 | 57.2 | 64.8 | 58.5 | 31.3 | 47.5 (+3.2) |
| + DPO | 55.4 | 61.4 | 74.8 | 66.9 | 32.6 | 53.6 (+9.3) |
| **CodeLlama-7b-hf** | 33.5 | 52.0 | 55.7 | 41.7 | 12.9 | 29.6 |
| + SFT | 29.5 | 47.6 | 58.9 | 38.5 | 16.1 | 31.3 (+1.7) |
| + DPO | 38.7 | 49.2 | 67.5 | 45.7 | 17.9 | 36.1 (+6.5) |
| **CodeQwen1.5-7B** | 51.8 | 72.2 | 70.0 | 70.1 | 49.7 | 61.1 |
| + SFT | 54.3 | 74.8 | 70.9 | 67.9 | 49.7 | 61.9 (+0.8) |
| + DPO | 55.5 | 75.4 | 72.5 | 66.9 | 45.7 | 61.1 (+0) |
| **starcoder2-15b** | 46.3 | 66.2 | 69.5 | 65.4 | 50.3 | 58.0 |
| + SFT | 51.6 | 69.2 | 72.0 | 68.9 | 51.7 | 61.3 (+3.3) |
| + DPO | 57.0 | 72.8 | 78.0 | 73.8 | 54.7 | 66.7 (+8.7 |
| **CodeLlama-13b-hf** | 37.8 | 62.4 | 76.8 | 60.5 | 18.4 | 39.6 |
| + SFT | 39.5 | 59.8 | 65.5 | 54.8 | 19.5 | 39.5 (-0.1) |
| + DPO | 49.1 | 64.4 | 78.6 | 60.0 | 29.0 | 50.1 (+10.6) |
| **deepseek-coder-33b-base** | 54.3 | 73.2 | 70.9 | 67.9 | 62.3 | 65.0 |
| + SFT | 58.1 | 74.8 | 61.8 | 58.0 | 47.1 | 58.7 (-6.3) |
| + DPO | **72.9** | **80.6** | 78.9 | **76.5** | 61.6 | **73.4** (+8.4) |
| **CodeLlama-34b-hf** | 48.2 | 65.4 | 77.7 | 63.7 | 32.4 | 52.4 |
| + SFT | 52.8 | 68.2 | 61.8 | 58.0 | 26.2 | 47.5 (-4.9) |
| + DPO | 65.9 | 75.2 | **83.9** | 68.4 | **63.2** | 70.6 (+18.2) |

**Code Efficiency.** Regarding the NL2Code task, once functional correctness has been assured, attention naturally pivots to enhancing code efficiency. As depicted in Table 3, we investigate code efficiency metrics across a spectrum of Code LLMs. Experiments demonstrate that DPO yields a stable enhancement in code efficiency from models exceeding 6.7B parameters. Notably, "deepseek-coder-33b-base" achieves the highest Beyond score of 66.47, marking a significant improvement of 17.94 over the vanilla model. In contrast, SFT detracts most Beyond scores from original models, suggesting that the plain SFT may not be a feasible strategy for enhancing code efficiency.

To investigate whether prompt engineering could offer a straightforward efficiency boost, we conducted an additional experiment using a specific prompt: *You are a coding expert. You can generate correct and fast code.* As outlined in Appendix Table 8, pre-trained code LLMs struggled to interpret the instruction, leading to decreased performance. Conversely, the instruction-tuned model "deepseek-coder-33b-instruct" demonstrated significant performance improvement, likely due to its training on Leetcode-style tasks, which enables it to effectively interpret the given instructions. By employing this simple prompt engineering technique, the Gap score is reduced from 10.6 to 8.8.

**Gap between Correctness and Efficiency.** Further analysis compares the Gap between Beyond and Pass. Since the ideal Beyond should be aligned with Pass (where the LLM-generated solution is correct and faster than all historical solutions), it shows how much the baseline method shrinks the gap between functional correctness and code efficiency. Our findings indicate that DPO substantially narrows Gap in models larger than 15B parameters. However, Gap tends to widen in smaller models under the same configuration. This implies that larger models possess a greater capacity to assimilate the nuanced knowledge to make strides in efficiency while retaining their functional correctness.

Table 3: Code efficiency (`Beyond`) evaluation results across three difficulty levels. **The bolded value** indicates the top performance for each metric, while the underlined values denote the most effective approaches among the original model and the baselines. In our experiment, we sample 5 solutions for each task to calculate *Beyond* score.

| Model name | Easy | Medium | Hard | Overall | Gap |
|---|---|---|---|---|---|
| **deepseek-coder-1.3b-base** | 47.97 | 39.77 | 19.26 | 35.62 | 9.85 |
| + SFT | 42.58 | 38.12 | 18.67 | 33.04 (-2.58) | 12.74 (+2.89) |
| + DPO | 46.91 | 42.27 | 16.78 | 35.21 (-0.41) | 9.64 (-0.21) |
| **starcoder2-3b** | 43.55 | 41.91 | 15.21 | 33.40 | 9.72 |
| + SFT | 44.64 | 42.10 | 15.72 | 34.01 (+0.61) | 14.04 (+4.31) |
| + DPO | 43.70 | 41.02 | 12.99 | 32.42 (-0.99) | 16.33 (+6.61) |
| **deepseek-coder-6.7b-base** | 48.80 | 51.16 | 45.11 | 48.29 | 16.40 |
| + SFT | 51.37 | 52.71 | 44.28 | 49.39 (+1.09) | 16.55 (+0.16) |
| + DPO | 56.25 | 52.35 | 40.62 | 49.70 (+1.41) | 18.73 (+2.34) |
| **starcoder2-7b** | 50.23 | 51.29 | 20.25 | 40.37 | 10.95 |
| + SFT | 42.21 | 44.02 | 21.09 | 35.61 (-4.77) | 15.80 (+4.84) |
| + DPO | 53.52 | 51.41 | 17.35 | 40.56 (+0.18) | 17.41 (+6.46) |
| **CodeLlama-7b-hf** | 42.55 | 30.99 | 8.88 | 27.45 | 9.27 |
| + SFT | 39.75 | 26.89 | 9.55 | 25.41 (-2.04) | 12.48 (+3.21) |
| + DPO | 54.14 | 34.48 | 11.10 | 33.29 (+5.84) | 10.46 (1.19) |
| **CodeQwen1.5-7B** | 51.11 | 53.56 | 39.03 | 47.78 | 15.35 |
| + SFT | 54.16 | 51.43 | 38.05 | 47.82 (0.04) | 14.91 (-0.43) |
| + DPO | 56.07 | 51.55 | 38.05 | 48.52 (0.74) | 13.12 (-2.22) |
| **starcoder2-15b** | 58.18 | 52.09 | 37.34 | 49.17 | 12.55 |
| + SFT | 53.54 | 52.77 | 37.73 | 47.92 (-1.25) | 16.22 (+3.67) |
| + DPO | 68.29 | 59.54 | 48.97 | 58.95 (9.78) | 10.81 (-1.74) |
| **CodeLlama-13b-hf** | 57.00 | 44.25 | 12.99 | 38.01 | 13.79 |
| + SFT | 44.95 | 39.96 | 13.55 | 32.70 (-5.31) | 13.78 (-0.01) |
| + DPO | 67.09 | 55.72 | 19.72 | 47.39 (9.38) | 8.47 (-5.32) |
| **deepseek-coder-33b-base** | 51.26 | 48.90 | 45.43 | 48.53 | 18.50 |
| + SFT | 40.33 | 37.75 | 36.82 | 38.32 (-10.21) | 17.30 (-1.20) |
| + DPO | 74.59 | 68.91 | **55.98** | **66.47** (+17.94) | **5.79** (-12.70) |
| **CodeLlama-34b-hf** | 56.28 | 48.21 | 22.96 | 42.40 | 15.49 |
| + SFT | 45.49 | 44.96 | 20.73 | 36.91 (-5.50) | 11.61 (-3.88) |
| + DPO | **78.55** | **60.95** | 51.94 | 63.94 (+21.54) | 8.01 (-7.47) |

## 4.5 Failure Analysis

Table 4 provides the error breakdown of where Code LLMs misstep during the *Mercury* evaluation:

(1) **Generation Errors** arise from syntactical issues. The common manifestations include *improper indentation*, *mismatched parentheses*, or *unexpected truncation*. Fine-tuning introduces additional knowledge for Code LLMs to adapt the Mercury convention, emphasizing standard indentation, concise code, and minimal comments. Therefore, both SFT and DPO generally reduced these errors, while they may lead to catastrophic forgetting in relatively smaller models, such as "deepseek-coder-1.3b-base".

(2) **Execution Errors** differ from Generation Errors because they occur after the code has been successfully loaded. These errors emerge as exceptions, which could stem from various issues, such as flawed code logic, execution timeouts, memory leakage, or sandbox interruption. We observe that SFT tends to aggravate these errors on most models, whereas DPO mitigates these errors successfully.

(3) **Test Case Errors** are the most prevalent errors where the code is executed without exceptions, but the output fails to align with the expectation. DPO demonstrates the suppression of these errors, especially in relatively large models, while SFT tends to increase the occurrence of these errors across nearly all models. This suggests that direct SFT may lead to catastrophic forgetting in vanilla models, diminishing their ability to generate functionally correct code. In contrast, DPO not only enhances code efficiency but also more reliably preserves functional correctness.

Table 4: The distribution of failure cases across *Code Generation*, *Code Execution*, and *Test Case* errors. E/M/H indicates Easy/Medium/Hard levels, respectively. We sample 5 solutions for each task, so there are $256 * 5 = 1280$ solutions in total for each model.

| Model Name | Code Generation | | | Code Execution | | | Test Case | | | Passed | | |
|---|---|---|---|---|---|---|---|---|---|---|---|---|
| | E | M | H | E | M | H | E | M | H | E | M | H |
| **deepseek-coder-1.3b-base** | 82 | 85 | 59 | 17 | 33 | 95 | 74 | 73 | 180 | 267 | 214 | 101 |
| + SFT | 106 | 104 | 37 | 16 | 8 | 63 | 59 | 76 | 225 | 259 | 217 | 110 |
| + DPO | 90 | 108 | 40 | 17 | 5 | 49 | 63 | 75 | 259 | 270 | 217 | 87 |
| **starcoder2-3b** | 107 | 102 | 33 | 35 | 26 | 107 | 51 | 66 | 201 | 247 | 211 | 94 |
| + SFT | 97 | 93 | 24 | 29 | 13 | 90 | 47 | 61 | 211 | 267 | 238 | 110 |
| + DPO | 79 | 75 | 11 | 30 | 14 | 87 | 56 | 69 | 235 | 275 | 247 | 102 |
| **deepseek-coder-6.7b-base** | 107 | 101 | 30 | 16 | 5 | 56 | 12 | 20 | 105 | 305 | 279 | 244 |
| + SFT | 105 | 100 | 25 | 17 | 6 | 61 | 14 | 10 | 98 | 304 | 289 | 251 |
| + DPO | 87 | 82 | 23 | 12 | 6 | 58 | 15 | 23 | 98 | 326 | 294 | 256 |
| **starcoder2-7b** | 107 | 101 | 22 | 21 | 9 | 74 | 32 | 45 | 212 | 280 | 250 | 127 |
| + SFT | 105 | 100 | 22 | 18 | 13 | 72 | 32 | 55 | 205 | 285 | 237 | 136 |
| + DPO | 90 | 90 | 21 | 10 | 11 | 61 | 11 | 33 | 211 | 329 | 271 | 142 |
| **CodeLlama-7b-hf** | 23 | 28 | 23 | 41 | 69 | 122 | 131 | 139 | 234 | 245 | 169 | 56 |
| + SFT | 11 | 9 | 17 | 44 | 72 | 112 | 126 | 168 | 236 | 259 | 156 | 70 |
| + DPO | 9 | 10 | 12 | 23 | 56 | 117 | 111 | 154 | 228 | 297 | 185 | 78 |
| **CodeQwen1.5-7B** | 105 | 100 | 18 | 1 | 4 | 44 | 26 | 17 | 157 | 308 | 284 | 216 |
| + SFT | 105 | 101 | 16 | 4 | 3 | 35 | 19 | 26 | 168 | 312 | 275 | 216 |
| + DPO | 98 | 96 | 26 | 5 | 8 | 35 | 18 | 30 | 175 | 319 | 271 | 199 |
| **starcoder2-15b** | 105 | 100 | 20 | 4 | 7 | 49 | 25 | 33 | 147 | 306 | 265 | 219 |
| + SFT | 104 | 100 | 18 | 3 | 3 | 56 | 16 | 23 | 136 | 317 | 279 | 225 |
| + DPO | 83 | 64 | 10 | 1 | 1 | 33 | 13 | 41 | 141 | 343 | 299 | 251 |
| **CodeLlama-13b-hf** | 10 | 14 | 28 | 19 | 41 | 99 | 73 | 105 | 228 | 338 | 245 | 80 |
| + SFT | 46 | 52 | 32 | 29 | 19 | 111 | 77 | 112 | 207 | 288 | 222 | 85 |
| + DPO | 24 | 9 | 24 | 10 | 12 | 100 | 60 | 141 | 185 | 346 | 243 | 126 |
| **deepseek-coder-33b-base** | 105 | 103 | 26 | 11 | 11 | 47 | 12 | 16 | 91 | 312 | 275 | 271 |
| + SFT | 69 | 78 | 27 | 27 | 26 | 65 | 72 | 66 | 138 | 272 | 235 | 205 |
| + DPO | 56 | 75 | 15 | 9 | 7 | 86 | 28 | 13 | 66 | 347 | 310 | 268 |
| **CodeLlama-34b-hf** | 22 | 35 | 50 | 28 | 55 | 84 | 48 | 57 | 160 | 342 | 258 | 141 |
| + SFT | 35 | 97 | 50 | 37 | 19 | 56 | 96 | 54 | 215 | 272 | 235 | 114 |
| + DPO | 4 | 12 | 10 | 26 | 76 | 30 | 41 | 40 | 120 | 369 | 277 | 275 |

## 5   Related Work

**NL2Code Generation** is the task of generating a computer program that satisfies given specifications. Initial approaches to converting natural language to code relied on rigid methods like probabilistic grammars and domain-specific languages, having limited flexibility and scalability [23, 13]. The advent of statistical models, such as n-grams and Hidden Markov models, attempted to overcome these limitations but struggled with modeling complexity and dependencies [35, 46]. The transformational impact of the Transformer model [47] and its subsequent application to NL2Code [34] led to the development of LLMs like Codex, which significantly improved the task's feasibility by utilizing extensive unlabelled data sets [9]. Follow-up LLMs such as AlphaCode [29], CodeGen [36], PaLM-Coder [11], and StarCoder [28] continued to advance this research field, exhibiting emergent abilities in coding and debugging that mirrored human programmers.

**NL2Code Correctness Evaluation** currently focuses on gauging the functional correctness of generated code. As a pioneer, CodeBLEU [41] adapts the BLEU [39] metric into code generation. However, given the abstract nature of programming languages, distinct code can express the equivalent semantics, prompting subsequent benchmarks to harness test case fuzzing instead of the similarity measurement. For example, HumanEval [9] and MBPP [3] consist of hand-written Python programming tasks and corresponding test cases. EvalPlus [1] enhances HumanEval by incorporating extensive auto-generated test cases, constructing a more rigorous benchmark HumanEval+ to evaluate the functional correctness of LLM synthesized code. On the note of enhancing language inclusiveness, ODEX [48] integrates multiple natural languages, while MBXP [2] extends the benchmarks to cater

to a variety of programming languages, promoting polyglot code generation evaluation. Recent benchmarks have also begun to consider more aspects beyond functional correctness. For instance, the benchmark DS-100 [26] dives deeply into the data analysis scenarios, and CodeGen [36] contributes a benchmark for multi-turn code generation. For security-oriented code generation, SecurityEval [44] offers a concentrating benchmark on mining the vulnerability of generated code. BigCodeBench [53] introduces more sophisticated instructions and diverse function calls to gauge the true programming capabilities of LLMs in realistic scenarios. LiveCodeBench [22] continuously updates its problem set, ensuring contamination-free functional correctness evaluations. One more work may be related to our work. AlphaCode [29] employs language models to generate solutions for competitive programming problems, but they do not focus on optimizing the solution performance.

**NL2Code Efficiency Evaluation** Evaluating code efficiency has long been a crucial topic in software engineering. With the advent of code generation models, it is gaining even more attention in the code LLM evaluation. As a pioneering effort, DeepPERF [16] employs a fine-tuned transformer to generate performance-enhancing patches for C# programs, evaluating the similarity between these generated patches and those created by developers. PIE [43] provides a benchmark suite for deterministically assessing the performance of C++ code within the Gem5 [7] environment. More recently, EFFIBENCH [20] constructed an efficiency benchmark using 1,000 Python problems from LeetCode. SUPERSONIC [10] introduces a compact sequence-to-sequence model designed to iteratively optimize code performance. All these studies employ the relative speedup metric to evaluate code efficiency gains.

## 6  Limitations

In this work, we measure code efficiency under the assumption that the code runtime is uniformly distributed. The simplification streamlines code efficiency evaluation via limited solution samples. However, the distribution of code runtime in real-world scenarios is more intricate, which may call for more solution samples to support more precise modeling. Additionally, the presence of data contamination during the model training phase compromises the precision of the Mercury benchmark to reflect the performance of tainted models [22]. To mitigate this issue, we will update our benchmark via our open-sourced data collection framework to import new tasks dynamically, thus laying the groundwork for more detailed investigations in subsequent studies.

## 7  Conclusion

In this work, we introduced Mercury, the first code efficiency benchmark for NL2Code evaluation. Unlike prior work that focused on functional correctness, our benchmark highlights the importance of code efficiency. By crafting dedicated test case generators and sampling ground-truth solutions across all difficulty levels from Leetcode, we have developed a comprehensive and rigorous Code LLM evaluation frame. We evaluated leading Code LLMs against benchmarks and found that even though these models are proficient in generating functionally correct code, there is still considerable space for code efficiency improvement. As Code LLMs become more widely used, code efficiency determines factual productivity, where Mercury can gauge the vital metric. As a commitment to ongoing research and to foster further innovation in this area, we have open-sourced the Mercury dataset collection framework, laying the groundwork for future advancements in the field.

### Acknowledgments and Disclosure of Funding

This research is supported by A*STAR, CISCO Systems (USA) Pte. Ltd and National University of Singapore under its Cisco-NUS Accelerated Digital Economy Corporate Laboratory (Award I21001E0002). The authors would also like to thank the anonymous reviewers whose comments and suggestions helped improve the presentation of this work.

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

# A Appendix

## A.1 Dataset Nutrition Labels

Table 5: Definitions of the fields within the Mercury dataset.

| Field Name | Definition |
|---|---|
| id | Task ID. |
| slug_name | Task name. |
| meta_info | The field accommodating the task description and submission statistics. |
| difficulty | The difficulty level of the task. |
| pretty_content | The field introduces the task description, examples, and constraints in pure text. |
| solutions | Samples of solutions extracted from actual past submissions. |
| prompt | The prompt of the solution. |
| entry_point | The nominative entry point of the solution. |
| generator_code | A function to generate test cases. |
| test_cases | A collection of generated test cases. |
| convert_online | A function to format test cases for online evaluation. |
| convert_offline | A function to format test cases for offline evaluation. |
| evaluate_offline | A function designed to evaluate solutions in an offline setting. |

## A.2 Mercury Data Distribution and Customized Data Structures

Except for all built-in Python data structures, Mercury imports another two structures to enhance the diversity and complexity as shown in Figure 4.

```python
class ListNode(object):
    def __init__(self, val=0, next=None):
        self.val = val
        self.next = next

class TreeNode(object):
    def __init__(self, val=0, left=None, right=None):
        self.val = val
        self.left = left
        self.right = right
```

Figure 4: Mercury supports two customized data structures: TreeNode and ListNode.

| Splits | Easy | Medium | Hard | Sum |
|---|---|---|---|---|
| Mercury-train | 446 | 968 | 219 | 1,633 |
| Mercury-eval | 88 | 81 | 87 | 256 |

Table 6: *Mercury-eval* encompasses 256 tasks, the difficulty level of which has been balanced for model evaluation. *Mercury-train* comprises the remaining 1,633 tasks for model training.

## A.3 Sandbox Details

**Time and Memory Limitation.** Each executed code within the sandbox is subject to certain constraints to ensure fair utilization of resources and to prevent any single code from monopolizing the system resource. Specifically, there are two primary constraints: a time limit and a memory limit. The time limit restricts how long the code can execute before being forcibly terminated, thereby ensuring that no infinite loops or excessively long computations negatively impact the availability of the sandbox. The memory limit caps the amount of RAM that a process can consume. This measure precludes a single code from exhausting the memory resources, which could lead to a denial of service for subsequent codes. In our experiment settings, the timeout limit is 30 seconds, and the memory limit is 2048 MB for each solution execution.

**IO Restriction.** To mitigate harmful activities such as unauthorized command execution or data exfiltration, the sandbox imposes strict Input/Output (IO) restrictions. These restrictions include limitations on reading from or writing to the disk and restrictions on the use of network sockets for sending or receiving data. By controlling the IO operations, the sandbox can prevent many common vulnerabilities and ensure that the code runs without interfering with other processes of the host system.

**Isolated File System.** The sandbox employs an isolated file system to provide a safe execution environment for the code. This means that the process running in the sandbox has its virtual file

system, which is separated from the host's file system. The isolated nature of this file system ensures that even if a process within the sandbox attempts to modify or delete files, these changes will not affect the host system or other sandboxes. It acts as a security layer, protecting the host from potential threats and maintaining the integrity of the overall system.

**System Libraries Redirection.**    To maintain a consistent and controlled environment, the sandbox redirects calls to system libraries to sandbox-specific versions. This is done to prevent code from using certain functions directly from the host's system libraries, which could result in unpredictable behavior or security vulnerabilities. The redirected libraries are often limited to a subset of functionalities deemed safe and necessary for executing programs within the sandbox, thus enforcing the security policies and ensuring that the running programs behave as expected.

**Single-threaded Evaluation.**    Single-threaded evaluation refers to executing code using a sole thread of execution, thereby simplifying resource management and timing assessments, and mitigating the intricacies linked with multi-threaded execution, such as synchronization issues, race conditions, and potential deadlocks. This mode of operation is especially important in testing environments where reproducibility and fairness are paramount, ensuring that each piece of code is evaluated using identical computational resources.

**Code Efficiency Measurement.**    Figure 5 shows the overview of the code execution pipeline. We gauge the *Solution Instantiation* and *Test Ease Evaluation* time spans as the execution runtime.

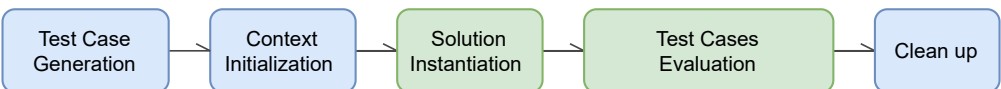

Figure 5: Sandbox Execution Pipeline. **1) Test Case Generation.** We first employ the corresponding test case generator for each task to produce a comprehensive set of test cases for the subsequent evaluation. **2) Context Initialization.** To prevent any unexpected code behavior, the sandbox environment is meticulously reinitialized for each new task. This phase ensures that all the common libraries required for executing the solution are loaded. **3) Solution Instantiation.** The solution under evaluation will be encapsulated as a *solution* class. **4) Test Case Evaluation.** Each test case the generator provides will be rigorously executed against the solution. A solution must successfully pass all the test cases to be deemed valid. **5) Clean up.** The final stage involves the sandbox dutifully clearing the namespace environment and the temporary directory. Mercury records the time consumed during the stage of Solution instantiation and Test Ease Evaluation as the primary metric for assessing code efficiency.

### A.4   DPO Experiment Details

**Dataset Construction.**    For every task problem $T^i$ in Mercury, we randomly selected two solutions from the task solution set $\{s^i_w, s^i_l\} \sim T^i_{solution}$, to construct the preference dataset $D = \{P^i, s^i_w, s^i_l\}$, where $p^i$ is the prompt, $s^i_w$ has a faster runtime than $s^i_l$.

**Model Initialization.**    RLHF [54] typically begins with a reference LLM $\pi_{ref}$. Here, we initialize $\pi_{ref}$ by maximizing the likelihood of faster code completions $(p, s_w) \sim D$, so that $\pi_{ref} = \arg\max_{\pi} E_{(p,s_w)\sim D}\left[\log \pi(s_w|p)\right]$. This procedure helps mitigate the distribution shift between the *true reference distribution* and $\pi_{ref}$.

**Optimization.**    We optimize the target LLM $\pi_{\theta}$ to minimize $\mathcal{L}_{DPO}$ for the given $\pi_{ref}$ and $D$ and desired hyperparameter $\beta$. The gradient with respect to the parameters $\theta$ can be written as $\nabla_{\theta}\mathcal{L}_{DPO}(\pi_{\theta}; \pi_{ref})$.

$$\mathcal{L}_{DPO}(\pi_{\theta}; \pi_{ref}) = -E_{(x,s_w,s_l)\sim D}\left[\log \alpha(\beta \log \frac{\pi_{\theta}(s_w|p)}{\pi_{ref}(s_w|p)}) - \log \frac{\pi_{\theta}(s_l|p)}{\pi_{ref}(s_l|p)})\right] \qquad (2)$$

$$\nabla_\theta \mathcal{L}_{DPO}(\pi_\theta; \pi_{ref}) =$$

$$- \beta E_{(p, s_w, s_l) \sim D} \left[ \underbrace{\alpha(\hat{r}_\theta(p, s_l) - \hat{r}_\theta(p, s_w))}_{\text{higher weight for wrong estimate}} \left[ \underbrace{\nabla_\theta \log \pi(s_w | p)}_{\text{increase likelihood of } s_w} - \underbrace{\nabla_\theta \log \pi(s_l | p)}_{\text{decrease likelihood of } s_l} \right] \right] \quad (3)$$

Intuitively, the gradient of the loss function $\mathcal{L}_{DPO}$ increases the likelihood of the preferred completions $s_w$ and decreases the likelihood of dis-preferred completions $s_l$, which are weighed by how much higher the implicit reward model $\hat{r}_\theta$ rates the dis-preferred completions, scaled by $\beta$, *i.e.*, how incorrectly the implicit reward model orders the completions, accounting for the strength of the KL constraint.

### A.5 External Libraries Utilized in Mercury

Raw LeetCode solutions typically commence without importing shared libraries. To avoid solution failure due to absent libraries, we proactively import the libraries listed in Figure 6 during the sandbox *Context Initialization* phase. Note that all these libraries are imported in a temporary namespace of which the sandbox controls code behaviors.

```python
exec('import re', namespace);
exec('import itertools', namespace);
exec('import collections', namespace);
exec('import heapq', namespace);
exec('import bisect', namespace);
exec('import string', namespace);
exec('import sys', namespace);
exec('import lctk', namespace);
exec('import functools', namespace);
exec('import math', namespace);
exec('import copy', namespace);
exec('import heapq', namespace);
exec('import sortedcontainers', namespace);

exec('from math import floor, ceil, factorial, sqrt, inf', namespace);
exec('from sys import maxsize, stdin', namespace);
exec('from bisect import bisect_left, bisect_right', namespace);
exec('from itertools import permutations, zip_longest', namespace);
exec('from heapq import heappush, heappop, heapify', namespace);
exec('from collections import deque, defaultdict, OrderedDict', namespace);
exec('from typing import List, Optional, Tuple', namespace);
exec('from functools import lru_cache, cache', namespace);
```

Figure 6: External Libraries Imported in Mercury Evaluate Framework.

### A.6 Model Details

Table 7: Model Scale and Corresponding HuggingFace Links

| Model Name | Model Scale | Link |
| --- | --- | --- |
| deepseek-coder-1.3b-base | 1.3B | https://huggingface.co/deepseek-ai/deepseek-coder-1.3b-base |
| starcoder2-3b | 3B | https://huggingface.co/bigcode/starcoder2-3b |
| deepseek-coder-6.7b-base | 6.7B | https://huggingface.co/deepseek-ai/deepseek-coder-6.7b-base |
| starcoder2-7b | 7B | https://huggingface.co/bigcode/starcoder2-7b |
| CodeLlama-7b-hf | 7B | https://huggingface.co/codellama/CodeLlama-7b-hf |
| CodeQwen1.5-7B | 7B | https://huggingface.co/Qwen/CodeQwen1.5-7B |
| CodeLlama-13b-hf | 13B | https://huggingface.co/codellama/CodeLlama-13b-hf |
| starcoder2-15b | 15B | https://huggingface.co/bigcode/starcoder2-15b |
| deepseek-coder-33b-base | 33B | https://huggingface.co/deepseek-ai/deepseek-coder-33b-base |
| CodeLlama-34b-hf | 34B | https://huggingface.co/codellama/CodeLlama-34b-hf |

## A.7 A Mercury Example

Given `n` non-negative integers representing an
elevation map where the width of each bar is `1`,
compute how much water it can trap after raining.
**Example**
Input: *height = [0,1,0,2,1,0,1,3,2,1,2,1]*
Output: *6*
Explanation: The above elevation map (black section) is represented by array
[0,1,0,2,1,0,1,3,2,1,2,1]. In this case, 6 units of rain water (blue section) are being trapped.

*Runtime: 125 ms*
```python
class Solution:
    def trap(self, height: List[int]) -> int:
        l,r = 0, len(height) -1

        total = 0
        maxLeft = height[l]
        maxRight = height[r]

        while l < r:
            if maxLeft < maxRight:
                l += 1
                maxLeft = max(maxLeft, height[l])
                total += maxLeft - height[l]
            else:
                r -= 1
                maxRight = max(maxRight, height[r])
                total += maxRight - height[r]

        return total
```

*Runtime: 600 ms*
```python
class Solution:
    def trap(self, height: List[int]) -> int:
        prev_greatest = []
        next_greatest = []
        total_tile_area = 0
        greatest = 0
        for i in range(len(height)):
            prev_greatest.append(greatest)
            greatest = max(greatest, height[i])
        greatest=0
        for i in range(len(height)):
            next_greatest.insert(0, greatest)
            greatest = max(greatest, height[len(height)-i-1])
        for i in range(1, len(height)-1):
            if min(next_greatest[i], prev_greatest[i]) > height[i]:
                total_tile_area += (abs(min(
                    next_greatest[i], prev_greatest[i]) - height[i]
                ))
        return total_tile_area
```

*Runtime: 2200 ms*
```python
class Solution:
    def trap(self, height: List[int]) -> int:
        total = 0
        maxLeft, maxRight = [height[0]], []
        currentMaxLeft = height[0]
        currentMaxRight = max(height[1:] or [0])
        for i in range(1, len(height)):
            maxLeft.append(currentMaxLeft)
            maxRight.append(currentMaxRight)
            if(height[i] > currentMaxLeft):
                currentMaxLeft = height[i]
            if(height[i] == currentMaxRight):
                currentMaxRight = max(height[i+1:] or [0])
        maxRight.append(0)
        for i in range(0, len(height)):
            current = min(maxLeft[i], maxRight[i]) - height[i]
            if current > 0: total += current
        return total
```

*Runtime: 4500 ms*
```python
class Solution:
    def trap(self, height: List[int]) -> int:
        if len(height) == 0 or len(height) == 1:
            return 0

        left_bound = height[0]
        right_bound = max(height[1:])
        water = 0

        for i in range(1, len(height)-1):
            right_bound = max(height[i+1:])

            if height[i] < left_bound and height[i]<right_bound:
                water += min(left_bound, right_bound) - height[i]
            elif height[i] >= left_bound:
                left_bound = height[i]

        return water
```

Figure 7: This case is drawn from the *Mercury-eval* benchmark. The upper block presents the problem statement with its example, while the subsequent portion exhibits the corresponding solutions. Although all solutions are functionally correct, they exhibit significant differences in runtimes.

## A.8 A HumanEval Example

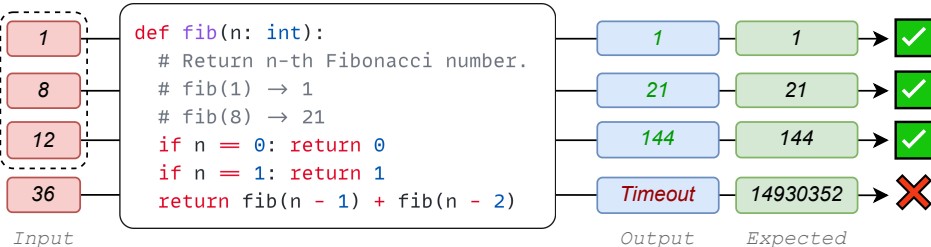

Figure 8: An HumanEval example of insufficient test cases. Even though the code passed all test cases in the dashed-line box, it remains vulnerable to timeout or stack overflow when subjected to a larger input.

## A.9 Prompts for Code Generation

To guarantee a fair comparison, we apply a unified one-shot prompt template for each pre-trained Code LLM. As displayed in Figure 9, the prompt template contains one shot example as well as three placeholders: *<task_content>*, *<code_starter>*, and *<code_completion>*.

```
1   ### Question
2   You are given a 0-indexed array of positive integers nums.
3   Find the number of triplets (i, j, k) that meet these conditions:
4       0 <= i < j < k < nums.length
5       nums[i], nums[j], and nums[k] are pairwise distinct.
6   Return the number of triplets that meet the conditions.
7
8   Example 1:
9       Input: nums = [4,4,2,4,3]
10      Output: 3
11      Explanation: The following triplets meet the conditions:
12      - (0, 2, 4) because 4 != 2 != 3
13      - (1, 2, 4) because 4 != 2 != 3
14      - (2, 3, 4) because 2 != 4 != 3
15      Since there are 3 triplets, we return 3.
16      Note that (2, 0, 4) is not a valid triplet because 2 > 0.
17
18  Example 2:
19      Input: nums = [1,1,1,1,1]
20      Output: 0
21      Explanation: No triplets meet the conditions so we return 0.
22
23  Constraints:
24  3 <= nums.length <= 100
25  1 <= nums[i] <= 1000
26
27  ### Code Prompt
28  class Solution(object):
29      def unequalTriplets(self, nums: List[int]) -> int:
30
31  ### Completion
32  class Solution(object):
33      def unequalTriplets(self, nums: List[int]) -> int:
34          ans = 0
35          n = len(a)
36          for i in range(n):
37              for j in range(i + 1, n):
38                  for k in range(j + 1, n):
39                      ans += len({a[i], a[j], a[k]}) == 3
40          return ans
41
42  ### Question
43  <task_content>
44
45  ### Code Prompt
46  <code_starter>
47
48  ### Completion
49  <code_completion>
```

Figure 9: Code Generation Prompts. Lines 1 to 40 are the one-shot example. In Mercury experiments, we feed the *pretty_content* field to the placeholder *<task_content>*, the *prompt* field to the placeholder *<code_starter>*, and the *solution* field to the placeholder *<code_completion>*

### A.10 Hardware-agnostic Evaluation

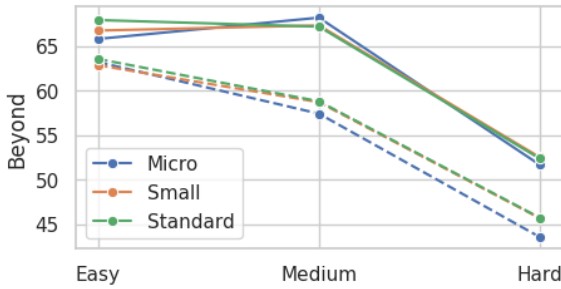

Figure 10: Beyond scores of 'deepseek-coder-33b' (solid line) and 'deepseek-coder-6.7b' (dashed line) across varied Intel Skylake CPU configurations. The results show that Beyond can remain consistent across different hardware configurations.

### A.11 Performance of Prompt Engineering on Mercury

Table 8: Performance of Prompt Engineering on Mercury.

| Model name | HumanEval | Pass | Beyond | Gap |
|---|---|---|---|---|
| **deepseek-coder-33b-base** | 54.3 | 67.2 | 48.5 | 18.7 |
| + Explicit instruction | 55.5 | 63.3 | 44.5 | 18.8 |
| **CodeLlama-34b-hf** | 48.2 | 57.8 | 42.4 | 15.4 |
| + Explicit Instruction | 47.6 | 53.5 | 36.9 | 16.6 |
| **deepseek-coder-33b-instruct** | 78.7 | 81.3 | 70.7 | 10.6 |
| + Explicit Instruction | 80.5 | 85.9 | 77.1 | 8.8 |

### A.12 Distribution of Bootstrapped Beyond Scores

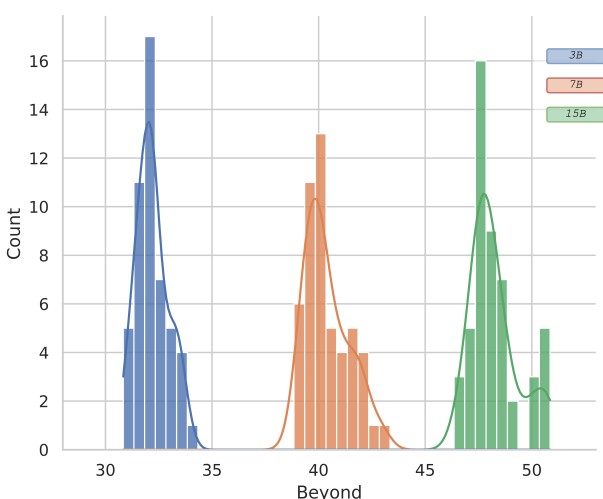

Figure 11: Bootstrapped Beyond Distribution. We evaluate 3B, 7B, and 15B Starcoder2[28] models using the *Mercury* benchmark. Each model was executed 50 times to ensure score robustness. The y-axis in the resulting histogram represents the frequency of observations within each bin.

### A.13  Dataset Metadata

The Mercury dataset is hosted on Huggingface: `https://huggingface.co/datasets/Elfsong/Mercury`. The Croissant Metadata can be found at `https://huggingface.co/api/datasets/Elfsong/Mercury/croissant`.

### A.14  Legal Compliance

In this study, we have curated a comprehensive dataset by gathering publicly accessible task descriptions and archived solutions from LeetCode (`https://leetcode.com/problemset/`). We have ensured that our collection process is strictly limited to tasks available in the free domain, intentionally excluding any content that falls under the paid services of the platform. We abide by Fair Use [37] (Section 107): *"the fair use of a copyrighted work, including such use by ... scholarship, or research, is not an infringement of copyright"*, where fair use is determined by *"the purpose and character of the use, including whether such use is of a commercial nature or is for nonprofit educational purposes"*. With the *Mercury* dataset, we emphasize its strictly non-commercial nature and underscore its purpose: to facilitate and advance academic research. The *Mercury* dataset is released under Creative Commons Attribution Non Commercial 4.0 [12].

