# OpenReview forum: "Mercury: A Code Efficiency Benchmark for Code Large Language Models"
_NeurIPS.cc/2024/Datasets_and_Benchmarks_Track — NeurIPS 2024 Track Datasets and Benchmarks Poster_

### Official Review · Reviewer_akjd · 2024-07-18

**Rating:** 8
**Confidence:** 4
**Correctness:** Yes
**Clarity:** Yes

**Review:**

**Quality:** The paper is well-written and easy to follow. The authors clearly outline their motivation and detail the process of constructing the dataset.

**Originality:** The paper presents a novel code efficiency benchmark and insights on enhancing code efficiency.

**Significance:** The work studies a timely and important topic. Code efficiency is a cornerstone in practical programming.

**Strengths:**

- This work introduces a comprehensive benchmark suite for evaluating the efficiency of LLM-generated code. The dataset and associated scripts are well-documented and publicly available.

- The datasets are categorized by difficulty, with each task including multiple real-world solutions. These solutions help create a runtime distribution that measures relative code efficiency.

- A thorough benchmark study of leading open-source LLMs is conducted. The experimental results compare the code efficiency performance between optimization baselines (SFT and DPO) and the original models.

**Additional Feedback:**

See Opportunities For Improvement.

**Documentation:**

Yes

**Limitations:**

The authors have adequately addressed the limitations in the paper.

**Opportunities For Improvement:**

- The paper primarily examines open-source LLMs. It would be interesting to see how commercial LLMs (such as those from OpenAI or Claude) perform on this benchmark.

- As the authors mentioned, all solutions are executed locally. However, varying system environments can affect runtime measurements. Could you elaborate on ensuring runtime results remain stable across different systems?

**Relation To Prior Work:**

Yes

**Summary And Contributions:**

The paper introduces Mercury,  a novel code efficiency benchmark for LLM code generation tasks. While existing benchmarks mainly focus on the functional correctness of generated code, this work bridges the gap in measuring their computational efficiency.

---

> ### Author Rebuttal · Authors · 2024-08-15
>
> Thank you for acknowledging our work. We would like to address your concerns as follows:
>
> > Q1: The paper primarily examines open-source LLMs. It would be interesting to see how commercial LLMs (such as those from OpenAI or Claude) perform on this benchmark.
>
> We're also curious about the performance of closed-source commercial LLMs on Mercury. The table below presents the Mercury performance ($Pass$ and $Beyond$) across closed-source models.
>
> | Model                  | HumanEval | Pass       | Beyond    | Gap      |
> |------------------------|-----------|------------|-----------|----------|
> | DeepSeek-Coder-V2 [1]  | **92.1%** | **90.6%**  | 68.2%     | 22.4%    |
> | GPT-4o [2]             | 91.5%     | 89.8%      | **71.6%** | 18.3%    |
> | GPT-4 [2]              | 89.6%     | 87.5%      | 62.5%     | 25.0%    |
> | GPT-3.5-Turbo [3]      | 70.1%     | 71.1%      | 48.1%     | 23.0%    |
> | Claude-3.5-Sonnet [4]  | 75.0%     | 82.4%      | 55.4%     | 27.1%    |
> | Claude-3-Haiku [4]     | 62.2%     | 66.8%      | 41.6%     | 25.2%    |
> | Gemini-1.5-pro [5]     | 76.2%     | 80.9%      | 64.4%     | **16.4%**|
> | Gemini-1.5-flash [5]   | 72.6%     | 78.5%      | 50.8%     | 27.7%    |
>
>
> > Q2: As the authors mentioned, all solutions are executed locally. However, varying system environments can affect runtime measurements. Could you elaborate on ensuring runtime results remain stable across different systems?
>
> Thank you for raising the point! The following measures have been taken to ensure stable and reliable efficiency scores across different systems:
>
> 1. **Invariant Nature of the Beyond Metric:** The $Beyond$ metric inherently mitigates runtime fluctuations caused by different system environments. While a high-performance system can indeed accelerate code execution and potentially shift the distribution, the normalized efficiency percentile remains consistent regardless of the underlying hardware and systems. We demonstrated that Mercury can provide a hardware-agnostic evaluation in Appendix Figure 10.
> 2. **Multiple Runs and Averaging:** Each generated solution is executed multiple times to obtain an average execution time. This approach helps smooth out variability and results in more stable and reliable runtime measurements.
> 3. **Isolated Sandboxing:** To minimize interference from other system processes, we execute code solutions within an isolated sandbox environment. This ensures that external factors do not affect the runtime results.
>
> We hope this explanation can address your concern. Should you have any further concerns, please do not hesitate to ask. Thank you once again for your thoughtful review and consideration!

---

> > ### Comment · Reviewer_akjd · 2024-08-30
> >
> > Thanks for your extensive response.
> >
> > My concerns have been addressed.

---

> > > ### Author Response · Authors · 2024-08-31
> > > **Thank you for your positive feedback!**
> > >
> > > Dear Reviewer akjd,
> > >
> > > Thank you for your positive feedback!
> > >
> > > We appreciate your recognition of our efforts and are delighted that our explanations and additional experiments have addressed your concerns. It means a lot to us. We will carefully incorporate the points we discussed in our final paper. Thank you once again!
> > >
> > > Best regards,
> > >
> > > The Authors

---

### Official Review · Reviewer_myqn · 2024-07-22
**A Code Efficiency Benchmark for Code LLMs**

**Rating:** 6
**Confidence:** 4
**Correctness:** Yes
**Clarity:** Yes

**Review:**

Mercury can be used to test LLM’s code generation with correctness and efficiency for a given problem. The benchmark collects four parts, the task description, test case generator, solutions to compete with, and prompt entry points for LLMs. The authors propose a metric "Beyond", which evaluates the score by the percentile of the code's running time between the provided solutions. This provides a uniform approach between platforms and has benefits for testing the LLMs' capabilities. Finally, the test case generator can provide a large number of test cases for evaluating the code.

However, there are some concerns about the benchmark. First of all, the considered metric tests  the solution of the LLM against the benchmark solution code's runtime. There can be biased solutions in the source (e.g., a very slow solution which could make all others score around 70~), which would make the comparison unfair for some problems. The next is the coverage of test cases generated by the generator. Unfortunately, it cannot be assured that GPT-4 creates a generator that considers all edge test cases.

**Strengths:**

- The authors focus on the neglected problem of testing the efficiency of generated codes, which is important in real-world computing tasks. The authors propose a framework that can be uniformly applied to various platforms and devices, by the use of metrics generated and tested on devices. This allows for a more generalized comparison between LLMs and provides more testing options for users.

- Evaluating 10 different LLMs and fine-tuning them with two different strategies (supervised fine-tuning and direct preference optimization) totaling 30 models demonstrates a thorough and rigorous analysis of the benchmark. Also, the provided results show that direct preference optimization can improve the LLMs to generate more efficient code.

- The authors provide the full stack for benchmarking on GitHub. Acquiring the benchmark and testing them on a model is easy and well-conducted.

**Additional Feedback:**

The following questions might help the reader to understand the paper better.

1. The algorithms and difficulty tags seem to be taken from LeetCode. Is there any verification on those tags?
2. What is the standard for labeling on the difficulties?
3. Do you think that overfitting might happen when finetuning the LLMs?

**Documentation:**

Yes

**Limitations:**

Yes

**Opportunities For Improvement:**

The authors should reinforce the test code generation and testing process. Currently, the test code generation process relies on GPT-4’s generation capability of test cases, and if the randomly generated test cases pass it doesn’t go under the verification process. There has to be a verification process for the test case generators.
Also, since the test code generator is created by GPT-4, the authors should show the experiment on GPT-4 and analyze the performance of GPT-4 to show that the test case generator is not biased in favor of GPT-4.

**Relation To Prior Work:**

Yes

**Summary And Contributions:**

This paper introduces “Mercury”, a benchmark for testing codes' efficiency. In this benchmark, the authors test the efficiency by the code running time in real devices. The authors ask a question on the current state of testing LLMs' ability to create high-quality code since they only test the functional correctness and do not consider the efficiency of the code. They also criticize the creation of testing functional correctness by creating the test cases manually. The benchmark is made to handle these issues, and the creation of this benchmark is conducted as follows:
1. Collect problems from LeetCode.
2. Filter the problems that have unique answers and a sufficient number of solution codes.
3. Make the test case generator by GPT-4, and test it on the original platform LeetCode.

---

> ### Author Rebuttal · Authors · 2024-08-15
>
> Thank you for your thoughtful review and insightful comments. We hereby address your concerns below.
>
> > Q1: First of all, the considered metric tests the solution of the LLM against the benchmark solution code's runtime. There can be biased solutions in the source (e.g., a very slow solution which could make all others score around 70~), which would make the comparison unfair for some problems.
>
> **Short Answer:** The extreme outliers indeed affect the runtime distribution. To avoid correct but very slow solutions (a very high upper bound), we have set a time limitation for each problem.
>
> **Long Answer:** We devised two types of $p_{n}^{k}$ to normalize the code efficiency, offering a trade-off in terms of runtime distribution density:
>
> * The first type $p_{k}^{n} = \frac{max(R^{n}) - clip(r_{k}^{n}, min(R^{n}), max(R^{n}))}{max(R^{n}) - min(R^{n})}$ utilized in our paper works better to sparse runtime distribution since it measures the distance between $r_k^n$ and the upper bound.
> * The second type $p_{n}^{k} = \frac{\Vert {r_j \mid r_k^n \leq r_j, r_j \in R_n}\Vert}{\Vert{R_n}\Vert}$ is designed for dense runtime distributions. This method helps prevent outliers but necessitates a larger number of solutions to adequately support the distribution. As we mentioned in the limitation, we are continously updating new solutions in the dataset.
>
> > Q2: The next is the coverage of test cases generated by the generator. Unfortunately, it cannot be assured that GPT-4 creates a generator that considers all edge test cases.
>
> Your concern about the coverage of all edge test cases is valid. While it is challenging to guarantee comprehensive coverage, this limitation does not compromise our measurements.
>
> Our assumption is that a more efficient solution will execute faster than a less efficient one across various individual test cases. Therefore, we need a statistically significant number of test cases rather than exhaustive coverage to achieve reliable efficiency measurements.
>
> As indicated in Q3, we have instructed GPT-4 to leverage the 'random' library to sample inputs within specified constraints. This approach enhances the diversity of test cases compared to manually crafted ones. Given that our main objective is to evaluate code efficiency, the current generator is adequate to provide a sufficient variety of test cases for meaningful analysis.
>
> > Q3: The authors should reinforce the test code generation and testing process. Currently, the test code generation process relies on GPT-4’s generation capability of test cases, and if the randomly generated test cases pass it doesn’t go under the verification process. There has to be a verification process for the test case generators.
>
> We have actually considered a verification process, but due to space limitations, we did not elaborate in the paper. Here are detailed steps to produce and verify test case generators (testers).
>
> * In our first attempt, we used GPT-4 to directly generate testers.
> While most generators were executable and generated expected test cases, we faced several limitations:
>     1. **Lack of Case Diversity:** Some generated testers hard-coded a list of test cases resembling example cases displayed in the problem description, which only provide limited and fixed cases.
>     2. **Hard Adhering to Constraints:** Testers often fail to obey input constraints accompanying most problems.
>     3. **Multiple Acceptable Answers:** For problems with multiple valid answers, handling all variants in the sandbox judge was cumbersome.
>     4. **Unusual Data Structures:** GPT-4 struggled with problems involving uncommon data structures, such as graphs (e.g., Leetcode 133) and standalone classes (e.g., Leetcode 208).
> * As briefly introduced in *Section Test Case Generator Line 123*, to mitigate limitations (1) and (2), we employed a few-shot prompt to instruct GPT-4 in generating more diverse and constraint-abiding testers. By presenting 'pretty_content' in the prompt, including problem descriptions, constraints, and the corresponding tester code, we encourage GPT-4 to use the ‘random’ library for sampling inputs within specified constraints. This approach significantly enhanced the diversity and validity of the generated test cases.
> * Regarding limitation (3), consider the example from Leetcode 2007. In this case, returning the correct array in any order is deemed an acceptable solution, which introduces extra complexity for the sandbox judge to handle these cases. Therefore, we ran all accepted solutions against the test cases produced by each tester and verified if the outcomes were consistent. To streamline the evaluation process, we have sifted out problems that could accept multiple valid solutions (mentioned in *Section Task Filters Line108*).
> * Following the exploration mentioned above, we also analyzed scenarios where GPT-4 fails to generate accurate testers. The primary errors fell upon limitation (4). Our analysis showed that GPT-4 struggles with handling unusual data structures, even with the few-shot prompting. Consequently, we excluded problems that involved unconventional data structures from our evaluation dataset (mentioned in *Section Task Filters Line 104*).
> * Finally, we manually verified and rectified all testers in the eval dataset to make sure they could generate valid test cases (mentioned in *Section Test Case Generator Line 127*).
> * We recognize that refining these details about test case generators will help the reader understand the paper better, so we will elaborate on the test code generation and testing process in our revision.

---

> > ### Author Rebuttal · Authors · 2024-08-15
> >
> > > Q4: Also, since the test code generator is created by GPT-4, the authors should show the experiment on GPT-4 and analyze the performance of GPT-4 to show that the test case generator is not biased in favor of GPT-4.
> >
> > Thank you for bringing up this insightful point! This problem may exist, but we think the bias is not serious because:
> > 1. These test case generators are specific to the problem, and the test cases are not directly generated by the model, so they will not directly introduce model bias.
> > 2. The table below presents the Mercury performance (Pass and Beyond) across closed-source models.  Since these models do not disclose the model details, we might analyze the performance of GPT-4 qualitatively.
> >     * **Model-wise (vertical) analysis:** The Mercury performance of GPT-4 falls between those of GPT-4o and GPT-3.5, which aligns with theirs performance on HumanEval.
> >     * **Benchmark-wise (horizontal) analysis:** We also list their HumanEval performance. The ranking is the same as Mercury,  showing that GPT-4 didn't get additional benefits (to make GPT-4 beyond GPT-4o) from the Mercury test cases.
> >
> > | Model                  | HumanEval | Pass       | Beyond    | Gap      |
> > |------------------------|-----------|------------|-----------|----------|
> > | DeepSeek-Coder-V2 [1]  | **92.1%** | **90.6%**  | 68.2%     | 22.4%    |
> > | GPT-4o [2]             | 91.5%     | 89.8%      | **71.6%** | 18.3%    |
> > | GPT-4 [2]              | 89.6%     | 87.5%      | 62.5%     | 25.0%    |
> > | GPT-3.5-Turbo [3]      | 70.1%     | 71.1%      | 48.1%     | 23.0%    |
> > | Claude-3.5-Sonnet [4]  | 75.0%     | 82.4%      | 55.4%     | 27.1%    |
> > | Claude-3-Haiku [4]     | 62.2%     | 66.8%      | 41.6%     | 25.2%    |
> > | Gemini-1.5-pro [5]     | 76.2%     | 80.9%      | 64.4%     | **16.4%**|
> > | Gemini-1.5-flash [5]   | 72.6%     | 78.5%      | 50.8%     | 27.7%    |
> >
> >
> > 3. To ensure current case generators are not biased in favor of GPT-4, the best way may be to produce testers by other LLMs (such as Claude and Gemini), and evaluate GPT-4 using this generators. However, as mentioned in Q3, this experiment may not be feasible to complete within the rebuttal period. Therefore, we manually get a set of test case generators to demonstrate their semantic equivalence, and to confirm that these generators are straightforward and do not contain any information beyond the problem requirements.
> >
> > ```python
> > # Problem: You are given a large integer represented as an integer array digits, where each digits[i] is the ith digit of the integer.
> > # The digits are ordered from most significant to least significant in left-to-right order.
> > # The large integer does not contain any leading 0's
> >
> > # GPT-4
> > import random
> > def generate_test_case():
> >     digits = [random.randint(1, 9)]
> >     for _ in range(random.randint(0, 99)):
> >         digits.append(random.randint(0, 9))
> >     return digits
> >
> > # Gemini Pro
> > import random
> > def generate_test_cases():
> >     digit_count = random.randint(1, 100)
> >     digits = [random.randint(0, 9) for _ in range(digit_count)]
> >     while digits[0] == 0:
> >         digits[0] = random.randint(0, 9)
> >     return digites
> >
> > # Claude Sonnet
> > import random
> > def generate_test_case():
> >     length = random.randint(1, 100)
> >     digits = [random.randint(1, 9)] + [random.randint(0, 9) for _ in range(length - 1)]
> >     return digits
> > ```
> >
> > > Q5: The algorithms and difficulty tags seem to be taken from LeetCode. Is there any verification on those tags? What is the standard for labeling on the difficulties?
> >
> > Yes, the problem and difficulty tags in our dataset are inherited from LeetCode. We did not implement additional verification for these tags. Since these tags are manually assigned and LeetCode has not disclosed the standards, it is important to note that LeetCode's difficulty ratings are subjective and can vary based on an individual's background and experience with specific topics.
> >
> > For human users, problems labeled 'Hard' are generally perceived as more challenging than those labeled 'Easy' or 'Medium'. However, this difficulty ranking may not accurately reflect the challenges posed to LLMs. Assessing the true difficulty level of problems for LLMs warrants a separate study. In our work, we retain the difficulty hierarchy from LeetCode to introduce more challenging (against human relatively) problems into our datasets and analyze the performance of LLMs across these varying difficulty levels.
> >
> > > Q6: Do you think that overfitting might happen when finetuning the LLMs?
> >
> > Fine-tuning models on LeetCode-style problems can make them more adept at this task, thereby potentially enhancing their general function-level code generation capabilities. For example, we can see the performance of HumanEval and MBPP are increasing along with fine-tuning. However, we strictly separated the Mercury training and test datasets. Therefore, our fine-tuned models do not suffer from overfitting.
> >
> > We hope this explanation can address your concern. Should you have any further concerns, please do not hesitate to ask. Thank you once again for your thoughtful review and consideration!
> >
> > [1] Zhu, Q., Guo, D., Shao, Z., Yang, D., Wang, P., Xu, R., ... & Liang, W. (2024). DeepSeek-Coder-V2: Breaking the Barrier of Closed-Source Models in Code Intelligence. arXiv preprint arXiv:2406.11931.
> >
> > [2] Achiam, J., Adler, S., Agarwal, S., Ahmad, L., Akkaya, I., Aleman, F. L., ... & McGrew, B. (2023). Gpt-4 technical report. arXiv preprint arXiv:2303.08774.
> >
> > [3] Brown, T., Mann, B., Ryder, N., Subbiah, M., Kaplan, J. D., Dhariwal, P., ... & Amodei, D. (2020). Language models are few-shot learners. Advances in neural information processing systems, 33, 1877-1901.
> >
> > [4] Team Anthropic. Anthropic Models. https://www.anthropic.com/
> >
> > [5] Team, G., Anil, R., Borgeaud, S., Wu, Y., Alayrac, J. B., Yu, J., ... & Ahn, J. (2023). Gemini: a family of highly capable multimodal models. arXiv preprint arXiv:2312.11805.

---

> > > ### Comment · Reviewer_myqn · 2024-08-22
> > >
> > > Thank you for clarifying my concerns about the paper. The additional metric is valid in the context you provided, but it may be different outside of that context. I suggest that you set an upper bound at a multiple of the fastest solution to give a hard cut on the solution set. I see that the test case generation was done with many iterations and that precautions were taken when generating the test cases. I understand that there are limitations to the GPTs generation capabilities and it was taken care of in the process.
> > > Also thank you also for the GPT4 experiments and the evaluation of the results. The results reassure that the test case generation process was not biased in a way that GPT recognizes. My concern about test case coverage for edge cases is indeed somewhat out of scope when considering efficiency, but it may be better if you could consider it at some point in the final paper. The reason for this is that problems may have a general solution that is viable for most cases, but the edge cases are not covered by that general solution. If the test case does not cover these cases, then it may not be a valid comparison.
> > > Overall, the direction of this research sounds interesting and I have adjusted my evaluation after reading the author's response.

---

> > > > ### Author Response · Authors · 2024-08-22
> > > > **Thank you so much for your response!**
> > > >
> > > > Dear Reviewer Myqn,
> > > >
> > > > Thank you so much for your response!
> > > >
> > > > We appreciate your recognition of our efforts and are delighted that our explanations and additional experiments have addressed your concerns. Your insights are invaluable in refining our work, and we are committed to carefully incorporating your suggestions in the final paper. Specifically, we will optimize the hard upper bound for the Beyond metric, include closed-source models as an auxiliary experiment, and provide a more detailed explanation of the test generator construction process. We believe these revisions will enhance our work. Thank you once again for your constructive feedback.
> > > >
> > > > Best regards,
> > > >
> > > > The Authors

---

### Official Review · Reviewer_yc6N · 2024-07-24
**Solid paper, but limited contribution to the field**

**Rating:** 6
**Confidence:** 4
**Correctness:** No issues with correctness.
**Clarity:** The paper is well written and logical…

**Review:**

This is a solid, self-contained paper that contributes a well-curated dataset of Python coding tasks taken from Leetcode.  Each task includes a description, test case generator and solution.  The test case generator is a nice contribution, although it is only tested on 24 inputs.

The key novelty of the paper is the Beyond metric for measuring code efficiency.  This is a straightforward calculation that computes an average over performance relative to provided reference solutions.  The reference solutions are needed as comparison points to account for variations in hardware performance.  This, of course, requires that all of the references solutions also be executed as part of the benchmark.

The Beyond metric has a couple of quirks -
* Scoring is going to be relative to a particular HW platform - different HW gives different scores, despite the fact that task descriptions are HW agnostic
* Scoring on a particular task is bounded by the best reference solution.  If the code generator devises a completely novel (superior) solution to a task, that is not awarded any additional credit owing to the clip function.  Similarly, an atrocious (but correct) solution is not penalized beyond the worst performing reference solution (or a failure).
* Because it is based on wall clock time and not on asymptotic performance, the metric has some bias toward smaller test cases which are more likely to perform inside the accepted range even the algorithm is asymptotically more costly.  That is, as the size of the inputs gets larger, the "real" efficiency of the solution is more likely to become evident.  That effect is somewhat masked by having only a few (presumably smaller) test cases.

The experiments were primarily noteworthy for the observation that DPO outperforms SFT.

Ultimately, this is a solid paper that makes a nice, but limited contribution to the field.  The dataset is generally useful for measuring CodeGen efficiency for Python, mostly due to the inclusion of a test case generator and multiple reference solutions.  The Beyond metric, which possibly useful, is a fairly naive method to measure efficiency.

**Strengths:**

This work provides a novel, curated dataset that can be used to measure generated code performance in Python.  The inclusion of multiple reference solutions of varying efficiency provides a comparison baseline for measuring the runtime of efficiency of generated solutions.  The dataset also includes a test case generator that is capable of providing more test cases than is usually included in benchmarks with a static set of test cases.

The Beyond metric leverages the comparison solutions to provide a simple measure of relative performance.  Experiments over 10 LLMs demonstrate the relative performance over different sized models.  The Beyond metric is a reasonable attempt to factor out the vagueness of runs over different hardware.

I appreciated the inclusion of the Gap analysis as a way to measure efficiency improvement relative to correctness.

**Additional Feedback:**

No additional feedback.

**Documentation:**

Link is provided (https://github.com/Elfsong/Mercury).  Documentation is sufficient.

**Ethics:**

No ethical concerns.

**Limitations:**

The limitations are adequately addressed.  However, see opportunities for improvement as described above.

**Opportunities For Improvement:**

* With regards to the experiments, the paper makes the claim that "model points located nearer to the diagonal line exhibit a more equitable balance between functionality and efficiency."  I think this statement is misleading.  In fact, the diagonal line represents perfect efficiency on all passing test cases - it's an upper bound on efficiency.  A higher pass rate is simply better.  There is no real tradeoff between Pass and Beyond - Beyond almost always increases as Pass increases (unless the new working solution is terrible).

* One thing I found interesting in the reported results was that there didn't seem to be any significant difference in the training improvement across the HumanEval, MBPP, Easy, Medium and Hard task sets.  The paper doesn't comment on this - was this expected?  Why?

**Relation To Prior Work:**

The paper contains only a brief summary of related work, focusing on other common code generation models and benchmarks.  Related work cites some relevant examples, but omits others (e.g., Phi-1 for codegen models).  Also, the paper uses HumanEval and MBPP as core benchmarks, but does not cite or use more recent benchmarks, including improvements to HumanEval (e.g., HumanEval+) that include more tests and correct known errors.  The related work section does not cover any prior effort to measure code efficiency.

**Summary And Contributions:**

This paper describes the Mercury benchmark, comprising -
* A dataset consisting of 1889 Python tasks at three (3) difficulty levels adapted from Leetcode
* A novel metric called Beyond that incorporates runtime performance (efficiency) as a metric
* Baseline results that include a conclusion that DPO surpasses SFT for improving code efficiency; this result was achieved by partitioning the dataset into training and evaluation subsets and then employing LoRA to adapt 10 different code models.

---

> ### Author Rebuttal · Authors · 2024-08-15
>
> > Q1: With regards to the experiments, the paper makes the claim that "model points located nearer to the diagonal line exhibit a more equitable balance between functionality and efficiency." I think this statement is misleading. In fact, the diagonal line represents perfect efficiency on all passing test cases - it's an upper bound on efficiency. A higher pass rate is simply better. There is no real tradeoff between Pass and Beyond - Beyond almost always increases as Pass increases (unless the new working solution is terrible).
>
> Thank you for your thorough reading and for pointing out this issue. We acknowledge that the original statement could indeed be misleading. We appreciate your suggestion and will revise the text accordingly. How does the following correction sound to you?
>
> "The diagonal line represents perfect efficiency to the corresponding correctness. Points closer to this line indicate better efficiency improvements relative to their correctness."
>
> > Q2: One thing I found interesting in the reported results was that there didn't seem to be any significant difference in the training improvement across the HumanEval, MBPP, Easy, Medium, and Hard task sets. The paper doesn't comment on this - was this expected? Why?
>
> The main reason is these experiments aim to investigate the innate capabilities of cutting-edge Code LLMs and their potential after baseline fine-tuning. Therefore, extensive parameter optimization and prompt engineering were not pursued.
>
> The functional correctness ($Pass$) evaluation results in Table 2 indicate that there are indeed no significant improvements for relatively small models (even performance drops in some cases). In contrast, substantial improvements are observed in relatively large models. These findings suggest that smaller models may struggle to integrate new knowledge while preserving their original functionality.
>
> > Q3: The paper contains only a brief summary of related work, focusing on other common code generation models and benchmarks. Related work cites some relevant examples, but omits others (e.g., Phi-1 for codegen models). Also, the paper uses HumanEval and MBPP as core benchmarks, but does not cite or use more recent benchmarks...
>
> We intend to broaden the scope of our related work by incorporating more leading code LLMs. Additionally, we will include a dedicated section in our revised manuscript to introduce more recent benchmarks and prior research efforts aimed at measuring and enhancing code efficiency. We have identified several relevant works and would appreciate any additional suggestions you might have.
>
> Regarding more recent benchmarks, EvalPlus [1] enhances HumanEval by incorporating extensive auto-generated test cases, constructing a more rigorous benchmark HumanEval+ to evaluate the functional correctness of LLM synthesized code. BigCodeBench [2] introduces more sophisticated instructions and diverse function calls to gauge the true programming capabilities of LLMs in realistic scenarios. Moreover, LiveCodeBench [3] continuously updates its problem set, ensuring contamination-free functional correctness evaluations.
>
> For the related work to measure code efficiency, PIE [5] offers a dataset of 77K C++ performance-improving pairs and a comprehensive benchmark suite to evaluate various code optimization strategies, including prompt engineering and fine-tuning. They design a relative efficiency metric $speedup = T_{new}/T_{original}$ and isolately evaluate the program in a gem5 environment. SUPERSONIC [4] introduces a small seq2seq model designed for iterative optimizing the code performance. Similar to PIE, SUPERSONIC employs the relative $speedup$ metric to gauge efficiency improvements.  DeepPERF [8] leverages a fine-tuned transformer to generate performance-improving patches for C# programs. It evaluates the similarity between the generated patches and those created by developers, while it does not quantitatively measure the performance improvement. One more work may be related to our work. AlphaCode [7] employs language models to generate solutions for competitive programming problems, but they do not focus on optimizing the solution performance.
>
> We hope this explanation can address your concern. Should you have any further concerns, please do not hesitate to ask. Thank you once again for your thoughtful review and consideration!
>
> [1] Liu, J., Xia, C. S., Wang, Y., & Zhang, L. (2024). Is your code generated by chatgpt really correct? rigorous evaluation of large language models for code generation. Advances in Neural Information Processing Systems, 36.
>
> [2] Zhuo, T. Y., Vu, M. C., Chim, J., Hu, H., Yu, W., Widyasari, R., ... & Von Werra, L. (2024). BigCodeBench: Benchmarking Code Generation with Diverse Function Calls and Complex Instructions. arXiv preprint arXiv:2406.15877.
>
> [3] Jain, N., Han, K., Gu, A., Li, W. D., Yan, F., Zhang, T., ... & Stoica, I. (2024). Livecodebench: Holistic and contamination free evaluation of large language models for code. arXiv preprint arXiv:2403.07974.
>
> [4] Chen, Z., Fang, S., & Monperrus, M. (2024). Supersonic: Learning to generate source code optimizations in C/C++. IEEE Transactions on Software Engineering.
>
> [5] Shypula, A., Madaan, A., Zeng, Y., Alon, U., Gardner, J., Hashemi, M., ... & Yazdanbakhsh, A. (2023). Learning performance-improving code edits. arXiv preprint arXiv:2302.07867.
>
> [6] Garg, S., Moghaddam, R. Z., & Sundaresan, N. (2023). Rapgen: An approach for fixing code inefficiencies in zero-shot. arXiv preprint arXiv:2306.17077.
>
> [7] Li, Y., Choi, D., Chung, J., Kushman, N., Schrittwieser, J., Leblond, R., ... & Vinyals, O. (2022). Competition-level code generation with alphacode. Science, 378(6624), 1092-1097.
>
> [8] Garg, S., Moghaddam, R. Z., Clement, C. B., Sundaresan, N., & Wu, C. (2022). Deepperf: A deep learning-based approach for improving software performance. arXiv preprint arXiv:2206.13619.

---

> > ### Comment · Reviewer_yc6N · 2024-08-29
> >
> > Your suggested revised language more accurately reflects the relationship between correctness and efficiency and is an improvement.  Thanks for making this change.
> >
> > Your other comments addressed my concerns on related work.

---

> > > ### Author Response · Authors · 2024-08-30
> > > **Thank you so much for your positive feedback!**
> > >
> > > Dear Reviewer yc6N,
> > >
> > > Thank you so much for your positive feedback!
> > >
> > > We are pleased to hear that the revisions have improved our work and addressed your concerns. It means a lot to us. We sincerely appreciate your suggestions and thoughtful review, which have been instrumental in enhancing the quality of this paper.
> > >
> > > Best regards,
> > >
> > > The Authors

---

### Official Review · Reviewer_2jdy · 2024-07-31
**Define and measure LLM code generation efficiency**

**Rating:** 6
**Confidence:** 3
**Correctness:** Correct!
**Clarity:** Clear to me.

**Review:**

- The Beyond metric you have proposed seems to be more of a relative measure, highly dependent on the quality of the collected solutions. What if the collection does not contain beset/worst solution? Is there a way to provide some absolute efficiency metrics?

- When constructing your dataset by filtering tasks, why must the tasks contain only the specified data structures? What is the rationale behind removing all other data structures?

-  In your efforts to improve code efficiency and the evaluation methodolodgy, have you tried using some simple prompting engineering as a baseline to evaluate the results? Would the same LLM generate different solutions?

**Strengths:**

- This paper proposes the first benchmark to comprehensively evaluate code efficiency in addition to functional correctness for LLMs. The Beyond metric effectively combines runtime efficiency with functional correctness.

- The Mercury benchmark includes 1,889 Python tasks with real-world efficiency baselines. This extensive dataset allows for thorough evaluation and comparison of different code generation models, enhancing the practical relevance of the benchmark.

- The paper features robust and extensive experiments, including comparisons between Direct Preference Optimization (DPO) and Supervised Fine-Tuning (SFT). The experimental setup and results are detailed and provide valuable insights into the performance and efficiency of various code generation models.

**Additional Feedback:**

N/A

**Documentation:**

Yes

**Limitations:**

See review above

**Opportunities For Improvement:**

See review above

**Relation To Prior Work:**

Yes

**Summary And Contributions:**

The paper introduces the first benchmark designed to evaluate both the functional correctness and computational efficiency of LLM for code generation. Mercury includes a dataset of 1,889 Python tasks, each with efficiency baselines, and introduces a novel metric called Beyond. Beyond provides a comprehensive assessment of both correctness and efficiency. The empirical results demonstrate that Direct Preference Optimization (DPO) significantly improves code efficiency compared to Supervised Fine-Tuning (SFT). Overall, Mercury offers a robust framework for advancing the evaluation and optimization of LLMs in generating efficient, correct code.

---

> ### Author Rebuttal · Authors · 2024-08-15
>
> Thank you for your valuable comments! We are more than happy to address your concerns below:
>
> > Q1: The Beyond metric you have proposed seems to be more of a relative measure, highly dependent on the quality of the collected solutions. What if the collection does not contain beset/worst solution? Is there a way to provide some absolute efficiency metrics?
>
> * You are correct that $Beyond$ depends on the collected solutions from LeetCode.
> * As a widely used online judge platform, LeetCode brings together a lot of attempts and explorations by programming enthusiasts. These solution data are precious for measuring and improving LLM code generation. We sampled these solutions from the LeetCode real submission distribution (from slow solutions to fast solutions), ensuring we can likely get the best and worst human solutions for each question.
> * One might consider using the raw code execution time as an absolute efficiency metric. However, this method suffers from variability introduced by system and hardware differences. For example, the same piece of code can run significantly faster on a high-performance machine than a low-performance one. To establish a model-wise ranking, users need to infer, execute, and count the execution time for all solutions across all models, which is impractical in real-world scenarios. Consequently, we propose the Beyond metric. It measures the percentile of code execution time and provides stable efficiency scores across different systems and hardware configurations. We demonstrated that Mercury can give a hardware-agnostic evaluation in Appendix Figure 10.
>
>
> > Q2: When constructing your dataset by filtering tasks, why must the tasks contain only the specified data structures? What is the rationale behind removing all other data structures?
>
> **Short Answer:** The rationale behind removing all other data structures is to minimize the unwarranted complexity of the sandbox judge.
>
> **Long Anser:** Since we need to handle the code output assessment in our sandbox judge, including uncommon data structures will import extra complexity, making the test case generation and evaluation process cumbersome. To streamline our approach, we exclude problems that involve uncommon data structures, such as Graphs (e.g., LeetCode 133) and some standalone classes (e.g., LeetCode 208). Fortunately, such problems constitute a small fraction (less than 5%) of the raw problem dataset. Our benchmark robustly supports *all built-in Python data structures*, as well as two commonly used data structures: *binary trees* and *linked lists*. Therefore, we have sufficient problems to evaluate the code efficiency metric.
>
> > Q3: In your efforts to improve code efficiency and the evaluation methodology, have you tried using some simple prompting engineering as a baseline to evaluate the results? Would the same LLM generate different solutions?
>
> To make a fair comparison, we consistently used the same prompt across our Vanilla, SFT, and DPO experiments. However, I believe that simple prompt engineering can provide an interesting baseline. To explore this, we conducted an additional experiment using an explicit instruction: *"You are a coding expert. You can generate correct and fast code."*
>
> | Model                                       | HumanEval | Pass  | Beyond | Gap        |
> |---------------------------------------------|-----------|-------|--------|------------|
> | deepseek-coder-33b-base                     | 54.3      | 67.2  | 48.5   | 18.7       |
> | + Explicit Instruction                      | 55.5      | 63.3  | 44.5   | 18.8       |
> | CodeLlama-34b-hf                            | 48.2      | 57.8  | 42.4   | 15.4       |
> | + Explicit Instruction                      | 47.6      | 53.5  | 36.9   | 16.6       |
> | deepseek-coder-33b-instruct                 | 78.7      | 81.3  | 70.7   | 10.6       |
> | + Explicit Instruction                      | 80.5      | 85.9  | 77.1   | 8.8        |
>
> As the experiment results shown in the above table, the pre-trained code LLMs 'deepseek-coder-33b-base' and 'CodeLlama-34b-hf' demonstrate difficulty in comprehending the instructions, even resulting in a performance decline. We hypothesize this is due to the lack of instruction tuning in these pre-trained models. Consequently, we evaluated an instruction-tuned model, 'deepseek-coder-33b-instruct'. Its performance markedly improves, presumably because it has been trained on Leetcode-style tasks and can effectively interpret the instructions provided. This simple prompt engineering technique reduced the Gap score from 10.6 to 8.8. We appreciate the suggestion that led to this insight, and we will incorporate this finding into our revised paper.
>
> Given different temperature settings, the same LLM does generate diverse solutions. To involve varying solutions in our Beyond score, we set the inference temperature as 0.2 and sampled 5 solutions for each problem (K=5).
>
> We hope this explanation can address your concern. Should you have any further concerns, please do not hesitate to ask. Thank you once again for your thoughtful review and consideration!

---

> > ### Comment · Reviewer_2jdy · 2024-08-26
> >
> > Can you elaborate more on why the including uncommon data structures will import extra complexity, or you can't be handled by your sandbox judge? Does it mean your solution cannot generalize to uncommon cases?
> >
> > Thanks for your additional experiments!

---

> > > ### Author Response · Authors · 2024-08-27
> > > **Thank you for your response!**
> > >
> > > Dear Reviewer 2jdy,
> > >
> > > Thank you for your response! We are more than happy to address your concern regarding additional data structures.
> > >
> > > We would like to clarify that our solution can generalize to uncommon cases, but doing so requires additional complexity, which arises from **the necessity of implementing serialization and deserialization functions for these uncommon data structures**. By providing these functions, our solution can seamlessly handle and generalize to uncommon cases that may involve atypical or complex data formats.
> > >
> > > For instance, consider Leetcode question 133 (Clone Graph), which defines a `GraphNode` structure as shown below. This question takes a `GraphNode` object as input and returns another `GraphNode` object as output.
> > >
> > > * Firstly, in order to persist test cases and compare them with the code output, we need to serialize these cases. Here, we outline two potential serialization methods below. Solution 1 utilizes an adjacency list, while Solution 2 employs an adjacency dictionary.
> > > * Next, we can import the data structure into the sandbox judge namespace. The consistency of serialization between the expected and actual outputs is then evaluated to determine functional correctness. A test case is considered as passed if `expected_output == actual_output`.
> > >
> > > In summary, **Our sandbox can generalize to uncommon cases. It supports any kind of customized data structure by providing the appropriate serialization and deserialization functions.** For the sake of simplicity, we have selected two commonly used customized data structures (binary tree and linked list) to demonstrate that our sandbox can support customized structures. We will elaborate on this point in our revision. We hope this explanation can address your concern. Thank you!
> > >
> > > Sincerely,
> > >
> > > The Authors
> > >
> > >
> > > ```python
> > > # GraphNode Data Struture
> > > class Node {
> > >     public int val;
> > >     public List<Node> neighbors;
> > > }
> > >
> > > # Example Graph
> > > 1 ---- 2
> > > |      |
> > > 4 ---- 3
> > >
> > > # Serialization Solution 1 (introduced in Leetcode):
> > > input = [[2,4],[1,3],[2,4],[1,3]]
> > > expected_output = [[2,4],[1,3],[2,4],[1,3]]
> > > actual_output = [[2,4],[1,3],[2,4],[1,3]]
> > >
> > > # Serialization Solution 2 (introduced in python 'lctk' library):
> > > input = {"$id":"1","neighbors":[{"$id":"2","neighbors":[{"$ref":"1"},{"$id":"3","neighbors":[{"$ref":"2"},{"$id":"4","neighbors":[{"$ref":"3"},{"$ref":"1"}],"val":4}],"val":3}],"val":2},{"$ref":"4"}],"val":1}
> > > ```

---

> > > > ### Author Response · Authors · 2024-09-01
> > > > **Thank you for your response!**
> > > >
> > > > Dear Reviewer 2jdy,
> > > >
> > > > We sincerely appreciate your thorough review and valuable suggestions. We will carefully incorporate your comments into the final paper, particularly by elaborating on the detailed processes regarding custom data structures and the prompt engineering experiment.
> > > >
> > > > As the Author-Review discussion period is nearing its end, we hope that our explanation adequately addresses your concerns. Thank you very much for your time and consideration!
> > > >
> > > > Best regards,
> > > >
> > > > The Authors

---

### Decision · Program_Chairs · 2024-09-26

**Decision:**

Accept (Poster)

**Comment:**

This paper creates and releases a new dataset of 1889 Python tasks gathered from LeetCode for 3 difficulty levels - with the goal of evaluating both the functional correctness and the computational efficiency of LLM generated code. Together with the tasks from LeetCode, the authors gather solutions submitted to LeetCode together with their runtime i.e. both slow and fast solutions. The authors then introduce a new metric called Beyond - which measures the runtime performance of the generated code, as a percentile, relative to the runtimes of the collected set of LeetCode submissions, by executing all the solutions and the generated code.

Using this dataset and the Beyond metric, the authors are able to establish that DPO significantly improves code efficiency as compared to SFT. Similar results can be established by using this dataset and the Beyond metric as a framework.

+ All reviewers find the contribution of this paper to be quite solid, well motivated, and the dataset to be well-curated. They also appreciate the test-code generator accompanied with every task in the dataset, which was generated by GPT-4.
+ Appreciate that the metric for measuring efficiency is HW agnostic and can be uniformly applied on various platforms.
+ Reviewers appreciate the breadth of LLM evaluation i.e. 10 LLMs, fine-tuned with SFT and DPO.
+ Reviewers are excited about the fact that DPO outperforms SFT on the metric of code efficiency.

- Reviewers are concerned about the quality of the test case generators, since they are produced by GPT-4, which can fail to cover corner cases. Also, the metrics may be biased in favor of GPT-4. These concerns have been addressed in the rebuttal and the authors have provided new results with GPT-4 and various other closed source models and the reviewers are satisfied.

Overall, this is a decent contribution to the area of hardware agnostic LLM evaluation for code efficiency, and all reviewers are positive about this paper, with most reviewers concerns addressed in the author rebuttals.